# Locally Private and Robust Multi-Armed Bandits

**Xingyu Zhou**
Wayne State University
xingyu.zhou@wayne.edu

**Wei Zhang** *
Texas A&M University
komo@tamu.edu

## Abstract

We study the interplay between local differential privacy (LDP) and robustness to Huber corruption and possibly heavy-tailed rewards in the context of multi-armed bandits (MABs). We consider two different practical settings: LDP-then-Corruption (LTC) where each user's locally private response might be further corrupted during the data collection process, and Corruption-then-LDP (CTL) where each user's raw data may be corrupted such that the LDP mechanism will only be applied to the corrupted data. To start with, we present the first tight characterization of the mean estimation error in high probability under both LTC and CTL settings. Leveraging this new result, we then present an almost tight characterization (up to log factor) of the minimax regret in online MABs and sub-optimality in offline MABs under both LTC and CTL settings, respectively. Our theoretical results in both settings are also corroborated by a set of systematic simulations. One key message in this paper is that LTC is a more difficult setting that leads to a worse performance guarantee compared to the CTL setting (in the minimax sense). Our sharp understanding of LTC and CTL also naturally allows us to give the first tight performance bounds for the most practical setting where corruption could happen both before and after the LDP mechanism. As an important by-product, we also give the first correct and tight regret bound for locally private and heavy-tailed online MABs, i.e., without Huber corruption, by identifying a fundamental flaw in the state-of-the-art.

## 1 Introduction

The Multi-Armed Bandit (MAB) problem (Berry & Fristedt, 1985) offers a fundamental approach for sequential decision-making under uncertainty based on only bandit feedback. Take online advertising as an illustrative example, where the advertising platform (i.e., the central learner) sequentially and adaptively displays ads (i.e., arm) based on users' reward feedback (e.g., engagement score) so as to maximize the cumulative rewards. In practice, several important factors have to be considered when designing real-world MAB algorithms, as illustrated below using online advertising.

**Privacy.** The raw engagement score (which is calculated based on clicks, purchases, and time spent viewing the ad, etc.) from a user's device may lead to privacy leakage. For instance, when the ad is about medicine on some rare or uncommon disease, a high engagement score might imply interest or association with the uncommon disease. Such privacy leakage may lead to unintended personal and social consequences as well as trust issues on the platform. One principled way to mitigate it is via local differential privacy (LDP) (Kasiviswanathan et al., 2011; Duchi et al., 2018), i.e., each user's device locally adds a suitable amount of noise (depending on the privacy mechanism and budget) to obfuscate the raw feedback before sending it out from the device (see the yellow region in Fig. 1).

**Robustness.** Another important factor in real-world scenarios is the robustness of MAB algorithms under both possibly heavy-tailed feedback and adversary corruption.

---

*Work done during research intern at Wayne State University

38th Conference on Neural Information Processing Systems (NeurIPS 2024).

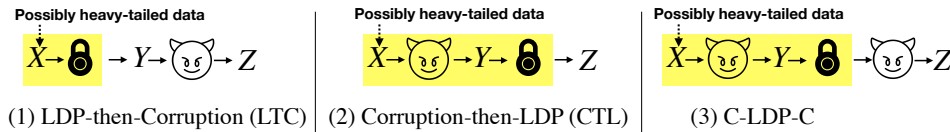

(1) LDP-then-Corruption (LTC) | (2) Corruption-then-LDP (CTL) | (3) C-LDP-C

Figure 1: The interplay between privacy and robustness (heavy-tailed data and corruption).

*Heavy-tailed feedback.* The engagement score in our example could often be heavy-tailed, i.e., non-negligible probabilities of observing extremely high values. This might happen due to some special events and seasons (e.g., Black Friday) or influencer interaction.

*Adversary corruption.* There could be malicious attacks on the engagement scores during the collection of users' feedback, e.g., with some probability, each score could be replaced by any *arbitrary* value, i.e., Huber corruption (Huber, 1964). On the other hand, corruption can also happen on each user's side before transmission, e.g., one could manipulate or spoof interactions to skew scores. Most practically, corruption can also happen both before and after the data transmission.

To tackle the above privacy and robustness issues in MABs, there has been a large related literature, which, however, mainly investigates the two issues in an isolated way (see Appendix B for details). Motivated by this, in this work, we are particularly interested in the following question:

> *Is there any interesting interplay between privacy and robustness in MABs?*

**Our contributions.** We give an affirmative answer to the above question by unveiling a fundamental interplay between privacy protection (in particular, local differential privacy (LDP)) and robustness under Huber corruption and heavy-tailedness. Our main message is a separation result between two MAB settings that differ in the order of privacy protection and corruption, i.e., LDP-then-corruption (LTC) vs. Corruption-then-LDP (CTL). That is, under LTC, corruption happens after LDP mechanism while under CTL, corruption happens before the LDP mechanism (see Fig. 1). To obtain our separation result for the two settings, we take the following principled approach:

**1.** We first study the mean estimation problem – a cornerstone step in the analysis of stochastic MABs – under both LTC and CTL settings. We give the first tight characterization of the estimation error in high probability, in terms of privacy budget, corruption level, and heavy-tailedness. Specifically, we first establish lower bounds on the minimax error rate in high probability and then propose a unified optimal algorithm that achieves matching worst-case upper bounds for both settings. The key observation here is that the mean estimation error under LTC is larger than that under CTL and moreover the gap becomes larger as the privacy requirement becomes stronger. Further, our sharp results on LTC and CTL also naturally enable us to give tight performance bounds for the most practical setting, C-LDP-C, where corruption happens both before and after LDP, see (3) in Fig. 1.

**2.** Leveraging the above tight mean estimation results, we then study both online MABs and offline MABs under both LTC and CTL. We present an almost tight characterization (up to log factor) of the corresponding minimax performances (i.e., regret in online MABs and sub-optimality in offline MABs) by deriving lower bounds and proposing almost optimal algorithms. As in mean estimation, there is a separation between LTC and CTL, i.e., LTC is a more difficult setting that leads to worse performance in the minimax sense, highlighting the interesting interplay between privacy and robustness in MABs. All of these results also allow us to easily handle the C-LDP-C setting.

**3.** Along the way, several results could be of independent interest. First, our optimal locally private and robust mean estimators can be applied to many other applications beyond MABs. Moreover, as an important by-product, we identify a fundamental flaw in the regret upper bound of state-of-the-art locally private online MABs with heavy tails (i.e., without corruption), and give the first correct one.

**Related Work.** We discuss the most relevant related work in the main body and relegate a detailed discussion to Appendix B. LDP with bounded/sub-Gaussian reward is first introduced to MABs in Ren et al. (2020) and later it was generalized to the heavy-tailed rewards (Tao et al., 2022). Robust MABs under Huber corruption have been recently studied in Kapoor et al. (2019); Mukherjee et al. (2021); Basu et al. (2022); Agrawal et al. (2023) while robust MABs concerning heavy-tailed reward date back to Bubeck et al. (2013). However, these work only study privacy and robustness separately. To the best of our knowledge, there are only two very recent work that consider privacy and robustness in MABs simultaneously. In Wu et al. (2023), the authors consider the central DP model where the raw non-private feedback received by the central learner can be first corrupted under Huber model. This is in sharp contrast to our local DP model, which is not only stronger but allows us to study

the order of corruption and privacy. In Charisopoulos et al. (2023), the authors study linear bandits (which includes MAB as a special case) under LDP and then Huber corruption (i.e., LTC setting). As will be discussed in Section 4, their regret bound is sub-optimal and worse than ours when reduced to the MAB case. Note that we also study the CTL setting, which in turn allows us to study the most practical setting C-LDP-C. Finally, our work is inspired by recent advances in (locally) private and robust mean estimation (Li et al., 2022b; Cheu et al., 2021; Chhor & Sentenac, 2023). Our key contributions are the first *high-probability* concentration bounds for both CTL and LTC settings.

## 2  Problem Setup

In this section, we formally introduce the three problems considered in this paper: mean estimation, online and offline MABs, under the constraints of both LDP and robustness (including heavy tails and Huber corruption). To start with, we introduce the privacy and corruption models.

**Definition 1** ($\varepsilon$-LDP, Duchi et al. (2018)). For a privacy parameter $\varepsilon \in [0, 1]$, the random variable $\widetilde{X}$ is an $\varepsilon$-*locally differentially private* view of $X$ via privacy channel/mechanism $Q$ if

$$\sup_{S \in \sigma(\widetilde{\mathcal{X}}), x, x' \in \mathcal{X}} \frac{Q(\widetilde{X} \in S \mid X = x)}{Q(\widetilde{X} \in S \mid X = x')} \le e^\varepsilon,$$

where $\sigma(\widetilde{\mathcal{X}})$ denotes an appropriate $\sigma$-field on $\widetilde{\mathcal{X}}$. In this case, we also say that the conditional distribution (privacy channel) $Q$ is an $\varepsilon$-LDP privacy mechanism. We write $\mathcal{Q}_\varepsilon$ as the set of all $\varepsilon$-LDP mechanisms (channels).

**Definition 2** ($\alpha$-Huber corruption, Huber (1964)). Given a parameter $\alpha \in [0, 1/2)$ and a distribution $D$ on inliers, the output distribution under $\alpha$-Huber model is $O = (1 - \alpha)D + \alpha E$. That is, a sample from $O$ returns a sample from $D$ with probability $1 - \alpha$ and otherwise returns a sample from some (unconstrained and unknown) corruption distribution $E$. We write $\mathcal{C}_\alpha(D)$ as the set of all possible $\alpha$-Huber corruptions (channels) of inlier distribution $D$.

With the two definitions in hand, we can introduce the two main settings in this paper: (i) LDP-then-Corruption (LTC) vs. (ii) Corruption-then-LDP (CTL), as also illustrated in Fig. 1.

**Definition 3** (LTC vs. CTL). We consider the following interplay between privacy and corruption.

(i) **LDP-then-Corruption (LTC):** Each user $i \in [n]$ first generates an $\varepsilon$-LDP view of raw data $X_i$. Then, the private data $Y_i$ from each device is independently corrupted by an $\alpha$-Huber channel that outputs $Z_i$ to the central analyzer/agent.

(ii) **Corruption-then-LDP (CTL):** Each user's raw data $X_i$ is first independently corrupted by an $\alpha$-Huber model. Then, the corrupted data $Y_i$ passes through an $\varepsilon$-LDP mechanism at each device that outputs $Z_i$ to the central analyzer/agent.

Under both settings, we aim to design $\varepsilon$-LDP mechanisms for user devices and central analyzers that ensure local privacy and robustness against $\alpha$-Huber corruption and heavy-tailed data distributions. The two settings also naturally enable us to study the most practical setting C-LDP-C.

**Mean estimation.** As in Duchi et al. (2018), given a real number $k > 1$, we consider the following class of possibly heavy-tailed distributions

$$\mathcal{P}_k := \{\text{distributions } P \text{ such that } \mathbb{E}_{X \sim P}[X] \in [-1, 1] \text{ and } \mathbb{E}_{X \sim P}[|X|^k] \le 1\}. \tag{1}$$

That is, $k$ controls the tail behavior of the distribution with smaller $k$ meaning heavier of the tails. Given any distribution $P \in \mathcal{P}_k$, our goal is to estimate its mean $\mu(P)$ as accurately as possible. In contrast to the standard case where the analyzer has access to $i.i.d$ samples $\{X_i\}_{i=1}^n$ from $P$, the analyzer in this paper now only observes samples $\{Z_i\}_{i=1}^n$ that are both private and corrupted view of $\{X_i\}_{i=1}^n$. Specifically, we are interested in the *high probability* error under our two different settings (LTC vs. CTL), as formally defined below.

**Definition 4** (Minimax mean estimation error rate). Given $\delta > 0$ and sample size $n > 0$, the minimax mean estimation error rate of the class $\mathcal{P}_k$ under $\varepsilon$-LDP and $\alpha$-Huber corruption is defined as follows

$$\phi_\delta^*(k, \varepsilon, \alpha, n) := \inf\{\phi > 0 \mid \inf_{Q \in \mathcal{Q}_\varepsilon} \inf_{\widehat{\mu}_n} \sup_{P \in \mathcal{P}_k} \sup_{C \in \mathcal{C}_\alpha(P)} \mathbb{P}[|\widehat{\mu}_n - \mu(P)| > \phi] \le \delta\}, \tag{2}$$

where $\widehat{\mu}_n$ is a measurable function of $\{Z_i\}_{i=1}^n$, i.e., private and corrupted view of $n$ $i.i.d$ samples $\{X_i\}_{i=1}^n$ from $P \in \mathcal{P}_k$ that pass through $\varepsilon$-LDP channel $Q$ and $\alpha$-Huber corruption channel $C$. We write $\phi_{\delta,\mathrm{LTC}}^*(k, \varepsilon, \alpha, n)$ and $\phi_{\delta,\mathrm{CTL}}^*(k, \varepsilon, \alpha, n)$ for the settings of LTC and CTL.

Intuitively speaking, $\phi_\delta^*$ represents the minimal error rate that any $\varepsilon$-LDP estimator can achieve with high probability $1 - \delta$ for all distributions $P \in \mathcal{P}_k$ and all $\alpha$-Huber corruption models, hence taking inf over $Q$ and $\widehat{\mu}_n$ and sup over distribution and corruption. Thus, the goal in our mean estimation problem is to design an optimal $\varepsilon$-LDP mechanism $Q^\star$ at each user's side and an optimal analyzer $\widehat{\mu}_n^\star$ at the central analyzer in order to attain the minimax mean estimate error rate in (2).

**Online MABs.** At each round $t \in [T]$, the central learner/analyzer chooses an action/arm $a_t \in [K]$ according to a policy $\pi$ and receives a reward sample $X_t$ that is drawn from some distribution $P_{a_t}$ with unknown mean $r(a_t) := \mu(P_{a_t})$. Here, the policy is $\pi = \{\pi_t\}_{t=1}^T$ and $\pi_{t+1}$ is a measurable function of the data received by the end of round $t$, i.e., for each $t \in [T]$, $\mathcal{D}_t = \{(a, X^{(a)}(t))\}_{a \in [K]}$ where $X^{(a)}(t) := \{X_1^{(a)}, \ldots, X_{N_a(t)}^{(a)}\}$ and $\sum_{a \in K} N_a(t) = t$. That is, for each round $t$, $X^{(a)}(t)$ groups together all $N_a(t)$ rewards from each arm $a \in [K]$ where $N_a(t)$ is the total number of times that arm $a$ has been pulled by time $t$. The goal in online MABs is to characterize the minimax *clean* regret under our LTC and CTL settings defined below.

**Definition 5** (Minimax clean regret). Let $\mathrm{MAB}(k) := \{\{P_a\}_{a \in K} \mid P_a \in \mathcal{P}_k\}$ be the class of $K$-armed MAB instances with inlier distributions for each arm in $\mathcal{P}_k$. Then, the minimax clean regret is defined as

$$R^*(k, \varepsilon, \alpha, T) := \inf_{Q \in \mathcal{Q}_\varepsilon} \inf_\pi \sup_{I \in \mathrm{MAB}(k)} \sup_{C \in \mathcal{C}_\alpha(I)} \mathbb{E}\left[T \cdot r(a^\star) - \sum_{t=1}^T r(a_t)\right], \tag{3}$$

where $a_{t+1}$ is a measurable function (via $\pi$) of private and corrupted dataset $\{(a, Z^{(a)}(t))\}_{a \in [K]}$. Here, for any arm $a \in [K]$ and $t \in [T]$, $Z^{(a)}(t) := \{Z_1^{(a)}, \ldots, Z_{N_a(t)}^{(a)}\}$ is the private and corrupted view of $N_a(t)$ samples of $P_a$ that pass through $\varepsilon$-LDP channel $Q$ and $\alpha$-Huber corruption channel $C$. We write $R_{\mathrm{LTC}}^*(k, \varepsilon, \alpha, T)$ and $R_{\mathrm{CTL}}^*(k, \varepsilon, \alpha, T)$ for the settings of LTC and CTL, respectively.

The goal in online MABs is to design an optimal $\varepsilon$-LDP mechanism $Q^\star$ and optimal learning policy $\pi^\star$ so as to attain the minimax clean regret in (3).

*Remark* 1. As standard in the literature (Wu et al., 2023; Chen et al., 2022; Niss & Tewari, 2020), $r(\cdot)$ in (3) is the mean of inlier distributions while the randomness in the expectation is generated by both privacy and corruption.

**Offline MABs.** In the offline case, the analyzer cannot interact with users and instead, it is given a batch pre-collected dataset $\mathcal{D} = \{(a_i, X_i)\}_{i=1}^N$ sampled from some joint distribution of a behavior policy $\pi$ and reward distributions $\{P_a\}_{a \in [K]}$. As in Rashidinejad et al. (2021), we assume a finite concentrability coefficient $\beta^\star$ such that $1/\pi(a^\star) \leq \beta^\star$, where $a^\star$ is the optimal arm that has the largest mean and $\beta^\star$ captures deviation between the behavior distribution $\pi$ and the distribution induced by the *optimal* policy. The goal here is to characterize the minimax sub-optimality under our LTC and CTL settings defined below.

**Definition 6** (Minimax sub-optimality). Let

$$\mathrm{MAB}(\beta^\star, k) := \{(\pi, \{P_a\}_{a \in K}) \mid P_a \in \mathcal{P}_k \text{ and } 1/\pi(a^\star) \leq \beta^\star\}$$

be the class of $K$-armed MAB instances with distributions in $\mathcal{P}_k$ and concentrability coefficient $\beta^\star$. Then, the minimax sub-optimality is defined as

$$\mathrm{SubOpt}^*(\beta^\star, k, \varepsilon, \alpha, N) := \inf_{Q \in \mathcal{Q}_\varepsilon} \inf_{\widehat{a}} \sup_{I \in \mathrm{MAB}(\beta^\star, k)} \sup_{C \in \mathcal{C}_\alpha(I)} \mathbb{E}\left[|r(a^\star) - r(\widehat{a})|\right], \tag{4}$$

where $\widehat{a}$ is a measurable function of private and corrupted dataset $\{(a, Z^{(a)})\}_{a \in [K]}$ and $Z^{(a)} := \{Z_1^{(a)}, \ldots, Z_{N_a}^{(a)}\}$ is the private and corrupted view of $N_a$ samples of $P_a$ that pass through $\varepsilon$-LDP channel $Q$ and $\alpha$-Huber corruption channel $C$. We write $\mathrm{SubOpt}_{\mathrm{LTC}}^*(\beta^\star, k, \varepsilon, \alpha, N)$ and $\mathrm{SubOpt}_{\mathrm{CTL}}^*(\beta^\star, k, \varepsilon, \alpha, N)$ for LTC and CTL, respectively.

We remark that we assume the batch data is collected by an $\varepsilon$-LDP mechanism that can be specified by the learner. Note that as in the standard case, we do not control the behavior policy $\pi$ other than a finite $\beta^\star$. The goal here is to design an optimal $\varepsilon$-LDP mechanism $Q^\star$ (which protects local privacy for any users offering batch data) and optimal offline learning algorithm $\widehat{a}^\star$.

## 3 Mean Estimation

We start with our first problem – mean estimation under privacy and robustness constraints. Our main result in this section is the following theorem that characterizes the minimax error rate (cf. Def. 4)

**Theorem 1** (Mean Estimation). *Given any fixed* $\delta \in (0, 1/2)^2$, $\varepsilon \in [0, 1]$, $\alpha \in [0, 1/2)$ *and* $k > 1$, *we have that for all large enough* $n$,

$$\phi_{\delta,\mathrm{LTC}}^*(k, \varepsilon, \alpha, n) = \Theta\left( \left(\frac{\alpha}{\varepsilon}\right)^{1-1/k} + \left(\frac{1}{\varepsilon}\sqrt{\frac{\log(1/\delta)}{n}}\right)^{1-1/k} \right),$$

$$\phi_{\delta,\mathrm{CTL}}^*(k, \varepsilon, \alpha, n) = \Theta\left( \alpha^{1-1/k} + \left(\frac{1}{\varepsilon}\sqrt{\frac{\log(1/\delta)}{n}}\right)^{1-1/k} \right).$$

*Remark* 2. To the best of our knowledge, this is the first high-probability concentration bound for mean estimation under both LTC and CTL, which tightly captures the dependence on the corruption level $\alpha$, privacy budget $\varepsilon$ and heavy-tail parameter $k$, simultaneously. It can be seen that for LTC setting, there is an additional $(1/\varepsilon)^{1-1/k}$ factor, which implies that introducing LDP guarantee first would make it more vulnerable to corruption/data manipulation attacks. Interestingly, for a fixed $\varepsilon$, this additional vulnerability due to LDP decreases as the tail becomes heavier, which offers additional insight into the interplay of privacy, heavy-tailedness, and robustness. Our LTC result also complements the result in Cheu et al. (2021), which considers the bounded case (i.e., $k = \infty$) under constant probability only rather than our high probability guarantee. On the other hand, for CTL, we note that the impact of corruption and privacy is *separable*. Our high probability bound for CTL complements the error bound in terms of mean-square error (MSE) only in Li et al. (2022b).

To establish Theorem 1, we first establish the following lower bounds, with full proof in Appendix E.

**Proposition 1** (Lower Bounds). *Given any fixed* $\delta \in (0, 1/2)$, $\varepsilon \in [0, 1]$, $\alpha \in [0, 1/2)$, $k > 1$ *and* $n$ *large enough, for all* $\varepsilon$-*LDP mechanism* $Q$ *and all estimator* $\widehat{\mu}_n$, *there exists a distribution* $P \in \mathcal{P}_k$ *and* $\alpha$-*Huber corruption channel* $C \in \mathcal{C}_\alpha(P)$ *such that with probability at least* $\delta$

*(i) For LTC:* $|\widehat{\mu}_n - \mu(P)| \geq \Omega\left( \left(\frac{\alpha}{\varepsilon}\right)^{1-1/k} + (\frac{1}{\varepsilon}\sqrt{\frac{\log(1/\delta)}{n}})^{1-1/k} \right)$,

*(ii) For CTL:* $|\widehat{\mu}_n - \mu(P)| \geq \Omega\left( \alpha^{1-1/k} + (\frac{1}{\varepsilon}\sqrt{\frac{\log(1/\delta)}{n}})^{1-1/k} \right)$,

*where recall that* $\widehat{\mu}_n$ *is a measurable function of* $\{Z_i\}_{i=1}^n$, *i.e., private and corrupted view of i.i.d samples* $\{X_i\}_{i=1}^n$ *from* $P \in \mathcal{P}_k$ *obtained from* $\varepsilon$-*LDP channel* $Q$ *and* $\alpha$-*Huber corruption channel* $C$.

***Proof sketch***. We provide a summary of the key steps in the proof. Essentially, we divide the proof into two parts. First, we consider the case without corruption and aim to establish the second term in the bound. To this end, we will leverage tools from information theory in an novel way, e.g., maximal coupling, strong data processing inequality of LDP, and Bretagnolle–Huber inequality between TV and KL distance. Then, we turn to give the first term related to corruption. To this end, we will leverage a folklore but important fact about Huber model. Roughly speaking, this fact says that given two inlier distributions $D_1$ and $D_2$ that satisfy $\mathrm{TV}(D_1, D_2) \leq O(\alpha)$, then after $\alpha$-Huber channel, one cannot distinguish between $D_1$ and $D_2$. Another important fact is that $\varepsilon$-LDP channel is a "contraction" channel in terms of TV distance, i.e., $\mathrm{TV}(M_1, M_2) \leq O(\varepsilon)\mathrm{TV}(P_1, P_2)$ where $M_1$, $M_2$ are induced marginals of $P_1, P_2$ after any $\varepsilon$-LDP channel. $\qquad\square$

**Key intuition behind the separation between LTC and CTL.** Building upon the above proof, one can immediately see that under the LTC setting, due to the "contraction" of LDP, one can choose two distributions that have a larger mean difference by a factor of $1/\varepsilon$, while still guaranteeing that after $\alpha$-Huber corruption, they are indistinguishable, *hence explaining the key difference of $1/\varepsilon$ between LTC and CTL.* We also provide another understanding of the separation from the attack perspective (see more details in Appendix A). The key idea here is that each single data attack in the LTC setting will lead to an additional $1/\varepsilon$ factor compared to CTL setting. This is mainly because any $\varepsilon$-LDP mechanism on binary data can be simulated by random response mechanism (Kairouz et al., 2015).

---

[2] We assume $\delta$ does not depends on $n$; otherwise, $\delta \in (\delta_{\min}, 1/2)$ where $\delta_{\min} = e^{-cn}$ for some $c > 0$.

We now turn to upper bounds, centering around the following key question: *Can we design a simple algorithm that can achieve optimal errors for all LTC, CTL, and even C-LDP-C in a unified way?* We give an affirmative answer via Algorithm 1. It consists of a local randomizer at each user's side and an analyzer at the central side. The task of $Q$ is to guarantee that its output is an $\varepsilon$-LDP view of its input. To this end, for each input $U_i$, it first truncates it into $\bar{U}_i$ using a properly chosen threshold $M$. Then, it converts the real number to binary data via random rounding. Next, it applies random response technique to generate the final output $\widetilde{U}_i$, i.e., with probability $\frac{e^\varepsilon}{e^\varepsilon+1}$, outputs a number of the same sign (with additional scaling for unbiasedness); otherwise flips the sign. Upon receiving the final input $\{Z_i\}_{i=1}^n$, the analyzer $\mathcal{A}$ first simply filters out the data if it is out of the bounded range and then returns the sample mean.

For LTC and CTL, the only difference in Algorithm 1 would be the truncation value $M$. The performance bounds for both settings under Algorithm 1 are given below. See Appendix F for proof.

---

**Algorithm 1** A Unified Algorithm

1: **Procedure:** $\varepsilon$-LDP mechanism $Q$
2: //Input: $U_i$, parameters: $M$, $\varepsilon$
3: //Output: private view $\widetilde{U}_i$
4: Truncate: $\bar{U}_i = U_i \mathbb{1}(|U_i| \leq M)$
5: Random rounding:
$$U_i' = \begin{cases} M & w.p. \ \frac{1+\bar{U}_i/M}{2} \\ -M & w.p. \ \frac{1-\bar{U}_i/M}{2} \end{cases}$$
6: Random response:
$$\widetilde{U}_i = \begin{cases} \frac{e^\varepsilon+1}{e^\varepsilon-1}U_i' & w.p. \ \frac{e^\varepsilon}{e^\varepsilon+1} \\ -\frac{e^\varepsilon+1}{e^\varepsilon-1}U_i' & w.p. \ \frac{1}{e^\varepsilon+1} \end{cases}$$
7: **Return** $\widetilde{U}_i$
8: **Procedure:** Analyzer $\mathcal{A}$
9: //Input: $\{Z_i\}_{i=1}^n$, parameters: $M$, $\varepsilon$
10: //Output: estimator $\widehat{\mu}_n$
11: **Return** $\widehat{\mu}_n = \frac{1}{n}\sum_{i=1}^n Z_i \mathbb{1}(|Z_i| \leq M \cdot \frac{e^\varepsilon+1}{e^\varepsilon-1})$

---

**Proposition 2** (Upper Bounds). *Given any fixed $\delta \in (0,1)$, $\varepsilon \in [0,1]$, $\alpha \in [0,1/2]$ and $k > 1$, for any distribution $P \in \mathcal{P}_k$ and any $\alpha$-Huber channel $C \in \mathcal{C}_\alpha$, Algorithm 1 satisfies that the mechanism $Q$ is $\varepsilon$-LDP and each returned estimate $\widehat{\mu}_n$ guarantees that with probability at least $1-\delta$*

*(i) For LTC:* $|\widehat{\mu}_n - \mu(P)| \leq O\left(\left(\frac{\alpha}{\varepsilon}\right)^{1-1/k} + \left(\frac{1}{\varepsilon}\sqrt{\frac{\log(1/\delta)}{n}}\right)^{1-1/k}\right)$,

*(ii) For CTL:* $|\widehat{\mu}_n - \mu(P)| \leq O\left(\alpha^{1-1/k} + \left(\frac{1}{\varepsilon}\sqrt{\frac{\log(1/\delta)}{n}}\right)^{1-1/k}\right)$,

*where (i) holds for* $M = \min\left\{\left(\frac{\varepsilon}{\alpha}\right)^{1/k}, \left(\frac{\varepsilon\sqrt{n}}{\sqrt{\log(1/\delta)}}\right)^{1/k}\right\}$ *and all* $n \geq 3\log(1/\delta)/\alpha$, *if* $\alpha > 0$;

*otherwise for all $n$ and $M = \left(\frac{\varepsilon\sqrt{n}}{\sqrt{\log(1/\delta)}}\right)^{1/k}$. (ii) holds for* $M = \min\left\{\left(\frac{1}{\alpha}\right)^{1/k}, \left(\frac{\varepsilon\sqrt{n}}{\sqrt{\log(1/\delta)}}\right)^{1/k}\right\}$

*and $n \geq 3\log(1/\delta)/\alpha$, if $\alpha > 0$; otherwise for $n \geq \log(1/\delta)$ and $M = \left(\frac{\varepsilon\sqrt{n}}{\sqrt{\log(1/\delta)}}\right)^{1/k}$.*

**Corruption-LDP-Corruption (C-LDP-C).** Our tight characterization of LTC and CTL immediately helps us understand the C-LDP-C setting, where corruption happens both before and after LDP. In particular, it is easy to see that the minimax lower bound for LTC would be a valid lower bound for the more difficult C-LDP-C setting. It turns out that this lower bound is also tight since it is matched by Algorithm 1 with the same parameter choice $M$ as in the LTC setting, see Appendix G.

**How to choose parameter $M$ in practice.** First, we note that for the bounded case ($k = \infty$), $M = 1$ across all three settings, independent of other parameters. This implies that Algorithm 1 can adaptively guarantee optimal minimax rates for LTC, CTL, and C-LDP-C without prior knowledge of the specific setting and other parameter like $\alpha$. Second, for certain applications, one may have prior knowledge of the underlying setting (see Appendix C.3). In this case, one can have a performance gain if it is under the CTL setting. Also, as mentioned above, we see that choosing the $M$ as in LTC can automatically help to handle the C-LDP-C setting. Finally, the dependence on $\varepsilon$ in $M$ is fine since it is a known privacy parameter while the dependence on the unknown parameter $\alpha$ is a little bit annoying. A quick practical fix is to use an estimated upper bound on $\alpha$. In theory, the story of whether one can remove it in our case is complicated, see the discussion in Appendix C.2.

*Remark* 3 (Burn-in period). Under Algorithm 1, when $\alpha > 0$, the concentration kicks in when the sample size $n$ is larger than a threshold. This type of burn-in period also exists in previous concentration results under the Huber model, though in different contexts (e.g., non-private case in Chen et al. (2022) or central model of DP in Wu et al. (2023)) or with different estimators (e.g., trimmed mean in Mukherjee et al. (2021)).

*Remark* 4 (Random response vs. Laplace mechanism). One may wonder if the standard Laplace mechanism can be applied in replace of the random response for $\varepsilon$-LDP in $Q$. The answer depends on the setting and the analyzer $\mathcal{A}$. For CTL, one can still derive a similar optimal concentration bound as in Proposition 2 by the concentration of Laplace noise. On the other hand, for LTC, simply replacing random response with Laplace mechanism in $Q$ will lead to an additional $\log(1/\alpha)$ factor. This aligns with the fact that truncation-based estimators even cannot achieve optimal mean estimation for Gaussians under corruption (Diakonikolas & Kane, 2023). The above discussion indicates another difference between LTC and CTL, i.e., the choice of $\varepsilon$-LDP mechanisms.

As two interesting applications of our mean estimation results, we will study both online MABs and offline MABs in the next two sections, highlighting again the sharp differences between LTC and CTL settings, in terms of regret and sub-optimality performance, respectively.

## 4 Online MABs

For online MABs, our main result is the following theorem that gives an almost tight characterization (up to log factor) of its minimax clean regret (cf. Def. 5) for both LTC and CTL settings.

**Theorem 2** (Online MABs). *Given any $\varepsilon \in [0,1]$, $\alpha \in [0,1/2)$ and $k > 1$, we have for all large enough $T$,*

$$R^*_{\delta,\text{LTC}}(k,\varepsilon,\alpha,T) = \widetilde{\Theta}\left(T \cdot \left(\frac{\alpha}{\varepsilon}\right)^{1-1/k} + T^{\frac{k+1}{2k}}\left(\frac{K}{\varepsilon^2}\right)^{\frac{k-1}{2k}}\right),$$

$$R^*_{\delta,\text{CTL}}(k,\varepsilon,\alpha,T) = \widetilde{\Theta}\left(T \cdot \alpha^{1-1/k} + T^{\frac{k+1}{2k}}\left(\frac{K}{\varepsilon^2}\right)^{\frac{k-1}{2k}}\right).$$

*Remark* 5. For both settings, due to corruption, the minimax clean regret (i.e., problem-independent regret) has a linear dependence on $T$, as in previous works under Huber corruption (Wu et al., 2023; Chen et al., 2022). The key here is to capture the tight factor in front of $T$, where the additional $1/\varepsilon$ factor in LTC again demonstrates the sharp difference between the two settings as in the mean estimation problem. As before, one can obtain the same rate for C-LDP-C from the LTC setting.

To prove the above theorem, we start with the corresponding lower bounds (see App. H for proof).

**Proposition 3** (Regret Lower bounds). *Let $\varepsilon \in [0,1]$, $\alpha \in [0,1/2)$, $k > 1$ and $T$ be large enough. Then, the minimax clean regrets satisfy the following results.*

*(i) LTC:* $R^*_{\text{LTC}}(k,\varepsilon,\alpha,T) \geq \Omega\left(T \cdot \left(\frac{\alpha}{\varepsilon}\right)^{1-1/k} + T^{\frac{k+1}{2k}}\left(\frac{K}{\varepsilon^2}\right)^{\frac{k-1}{2k}}\right);$

*(ii) CTL:* $R^*_{\text{CTL}}(k,\varepsilon,\alpha,T) \geq \Omega\left(T \cdot \alpha^{1-1/k} + T^{\frac{k+1}{2k}}\left(\frac{K}{\varepsilon^2}\right)^{\frac{k-1}{2k}}\right).$

**Comparisons to related work.** We first remark that Tao et al. (2022) studied a similar case but without corruption (i.e., $\alpha = 0$) and established a lower bound on the order of $\Omega\left(\left(\frac{K}{\varepsilon^2}\right)^{1-1/k}T^{1/k}\right)$ (for $k \in (1,2]$ when adapted to our setting), which is weaker concerning $T$ compared to our lower bound. In Tao et al. (2022), the authors also claimed to achieve their lower bound via some arm-elimination algorithm, which now becomes *ungrounded* given our tighter lower bound. That is, since for a large enough $T$, our lower bound is even larger than their upper bound for fixed $\varepsilon$, $k$ and $K$ (e.g., $T^{3/4}$ vs. $\sqrt{T}$ for $k = 2$, see further discussion in Appendix C.3). Another recent work Wu et al. (2023) also studies online MABs with both privacy and Huber corruption but under the *weaker* central model of DP. In particular, the true reward from each user may be first corrupted before being observed by the central learner, who is then responsible for taking care of privacy guarantees. That is, the central learner has access to users' raw (corrupted) data rather than only a private view of data as in our LDP case. Under this strictly weaker privacy model, Wu et al. (2023) establish the following

lower bound on the minimax clean regret: $\Omega\left(\sqrt{KT} + (K/\varepsilon)^{1-\frac{1}{k}} T^{\frac{1}{k}} + T\alpha^{1-\frac{1}{k}}\right)$. Compared to our CTL setting, one can see that our stronger LDP privacy incurs a larger privacy cost.

Now, let us turn to our proposed algorithm (i.e., Algorithm 2) for achieving matching regret upper bounds (up to log factor). Algorithm 2 is a variant of upper confidence bound (UCB)-based algorithm (cf. Auer et al. (2002)), which computes the UCB index for each arm at each round $t \in [T]$ and then selects the one with the highest UCB, i.e., optimism in the face of uncertainty. To construct a valid UCB, we resort to our mean estimation results in the last section. In particular, we will need Algorithm 1 to compute the private and robust sample mean $\widehat{\mu}_{a,N_a(t)}(t)$ for each arm $a \in [K]$ at each round $t$, where $N_a(t)$ be the number of pulls of arm $i$ by the beginning of time $t$. Then, the bonus term (i.e., radius of the confidence bound) $\beta_a(t)$ comes from the high probability mean estimation error established in Proposition 2. Note that due to burn-in period of the concentration results, Algorithm 2 has an additional exploration period to guarantee that the number of arm pulls is larger than a threshold (line 4). The following proposition formally states the regret guarantees of Algorithm 2 with the proof given in Appendix I.

---

**Algorithm 2** Online MABs under LTC and CTL

1: **Input:** private and robust mean estimator $\widehat{\mu}_n(k, \varepsilon, \alpha, \delta)$ in Algorithm 1, constant $c$
2: **Initialize:** For each $a \in [K]$, $\widehat{\mu}_{a,s}(t)$ is the estimate $\widehat{\mu}_s(k, \varepsilon, \alpha, t^{-4})$ based on the first $s$ observed values of $Z_{a,1}, \ldots, Z_{a,s}$ of the rewards for arm $a$;
3: **for** $t \in [T]$ **do**
4:    **if** $\exists a \in [K], N_a(t) \leq 6\log(t)/\alpha$ **then**
5:      $a_t = a$
6:    **else**
7:      Let $\gamma_a(t) := \left(\frac{1}{\varepsilon}\sqrt{\frac{\log(t^4)}{N_a(t)}}\right)^{1-1/k}$
8:

$$\beta_a(t) = \begin{cases} c\left(\frac{\alpha}{\varepsilon}\right)^{1-1/k} + c\gamma_a(t) & \text{LTC} \\ c\alpha^{1-1/k} + c\gamma_a(t) & \text{CTL} \end{cases}$$

9:      Let $\text{UCB}_a(t) = \widehat{\mu}_{a,N_a(t)}(t) + \beta_a(t)$
10:     $a_t = \operatorname{argmax}_{a\in[K]} \text{UCB}_a(t)$
11:    **end if**
12: **end for**

---

**Proposition 4** (Regret Upper Bounds). *Let $\varepsilon \in [0,1]$, $\alpha \in (0, 1/2)$, $k > 1$ and $T$ be large enough. Then, for any $1/2 > \bar{\alpha} \geq \alpha$, the expected clean regret of Algorithm 2 satisfies the following guarantees.*

*(i) LTC:* $R_{\text{LTC}}(k, \varepsilon, \alpha, T) \leq O\left(T\left(\frac{\bar{\alpha}}{\varepsilon}\right)^{1-1/k} + \left(\frac{K\log T}{\varepsilon^2}\right)^{\frac{k-1}{2k}} T^{\frac{k+1}{2k}} + \frac{K\log T}{\bar{\alpha}}\right)$;

*(ii) CTL:* $R_{\text{CTL}}(k, \varepsilon, \alpha, T) \leq O\left(T\bar{\alpha}^{1-1/k} + \left(\frac{K\log T}{\varepsilon^2}\right)^{\frac{k-1}{2k}} T^{\frac{k+1}{2k}} + \frac{K\log T}{\bar{\alpha}}\right)$.

**Comparisons to related work.** First, for $\alpha = 0$, our result with a direct modification of the burn-in period gives a regret bound that only has the term $O\left(\left(\frac{K\log T}{\varepsilon^2}\right)^{\frac{k-1}{2k}} T^{\frac{k+1}{2k}}\right)$. This is the first *correct* regret bound for locally private heavy-tailed MABs, i.e., without corruption, fixing the aforementioned issue in the state-of-the-art in Tao et al. (2022) (see more discussions in Appendix C.3). Second, it is worth comparing our result to a recent similar result in Charisopoulos et al. (2023), where the authors present regret for linear bandits under LTC setting. Their result is worse than ours when reduced to MAB with bounded rewards, as the scaling with respect to $\alpha$ is $\sqrt{\alpha}$ in the first linear term rather than our $\alpha$. Another minor difference is that our algorithm is anytime while their algorithm is not.

**Other extensions.** Although we mainly focus on minimax regret (i.e., problem-independent bound) in this paper, under some conditions of corruption level and the minimum mean gap, Algorithm 2 is also able to offer some problem-dependent bounds (see Appendix I). In the case that the corruption parameter $\alpha$ is very small but not equal to zero, one can tune the choice of $\bar{\alpha}$ (hence truncation threshold $M$) to balance the first and third terms in the bound. Similar comments and observations have been made in related work as in Chen et al. (2022); Wu et al. (2023).

## 5 Offline MABs

In this section, we study offline MABs as another application of our high probability mean estimation results developed in Section 3. We establish both lower bounds and almost matching upper bounds for locally private offline MABs with corruptions. To the best of our knowledge, this is the first result on offline MABs with heavy-tailed rewards, even without privacy and corruption.

**Proposition 5** (Sub-optimality Lower Bounds). *Let $\varepsilon \in [0,1]$, $\alpha \in [0, 1/2)$, $k > 1$ and $N$ be large enough. Then, for $\beta^\star \geq 2$, the minimax expected sub-optimality satisfies the following results.*

*(i) LTC:* $\mathrm{SubOpt}^*_{\mathrm{LTC}}(\beta^\star, k, \varepsilon, \alpha, N) \geq \Omega\left( \left( \left(\frac{\alpha}{\varepsilon}\right)^{1-1/k} + \left(\frac{1}{\varepsilon}\sqrt{\frac{\beta^\star}{N}}\right) \right)^{1-1/k} \right);$

*(ii) CTL:* $\mathrm{SubOpt}^*_{\mathrm{CTL}}(\beta^\star, k, \varepsilon, \alpha, N) \geq \Omega\left( \alpha^{1-1/k} + \left(\frac{1}{\varepsilon}\sqrt{\frac{\beta^\star}{N}}\right)^{1-1/k} \right);$

Now, let us turn to our proposed algorithm, which is able to achieve a matching expected sub-optimality (up to log factor) for both LTC and CTL settings. Our algorithm is a simple variant of the classic Lower Confidence Bound (LCB)-based algorithm as in Rashidinejad et al. (2021), i.e., pessimism in the offline setting. The key difference compared to Rashidinejad et al. (2021) is our new private and robust estimator (line 8) and penalty term (line 10), which come from our high probability mean estimation error. Another modification is due to our burn-in period of concentration result (line 4). Putting all of these together, Algorithm 3 is able to achieve the following guarantees on the expected sub-optimality, which almost matches the lower bound in Proposition 5. See the App. K and J for proofs of the upper and lower bounds.

**Proposition 6** (Sub-optimality Upper Bounds). *Let $\varepsilon \in [0,1]$, $\alpha \in (0, 1/2)$, $k > 1$ and $\delta = 1/N$. Then, for all finite $\beta^\star \geq 1$ and large enough $N$ and $N_{a^\star} \geq 3\log(1/\delta)/\alpha$, the expected sub-optimality of Algorithm 3 satisfies*

---

**Algorithm 3** Offline MABs under LTC and CTL

1: **Input:** Offline data $\mathcal{D} = \{(a, Z^{(a)})\}_{a \in [K]}$, mean estimator $\widehat{\mu}_n(k, \varepsilon, \alpha, \delta)$ in Algorithm 1, positive constant $c$
2: **Initialize:** $N_a = |Z^{(a)}|$ for all $a \in [K]$, i.e., number of pulls for arm $a$ in $\mathcal{D}$
3: **for** $a \in [K]$ **do**
4:     **if** $N_a < 3\log(1/\delta)/\alpha$ **then**
5:         Set the empirical mean reward $\widehat{r}(a) = 0$
6:         Set the penalty $b(a) = 1$
7:     **else**
8:         $\widehat{r}(a) = \widehat{\mu}_{N_a}(k, \varepsilon, \alpha, \delta)$
9:         Define $\gamma = \left( \frac{1}{\varepsilon}\sqrt{\frac{\log(2K/\delta)}{N_a}} \right)^{1-1/k}$
10:         $b(a) = \begin{cases} c\left(\frac{\alpha}{\varepsilon}\right)^{1-1/k} + c\gamma & \text{for LTC} \\ c\alpha^{1-1/k} + c\gamma & \text{for CTL} \end{cases}$
11:     **end if**
12: **end for**
13: **Return** $\widehat{a} = \arg\max_{a \in [K]} \widehat{r}(a) - b(a)$

---

*(i) LTC:* $\mathrm{SubOpt}_{\mathrm{LTC}}(\beta^\star, k, \varepsilon, \alpha, N) \leq O\left( \left(\frac{\alpha}{\varepsilon}\right)^{1-1/k} + \left(\frac{1}{\varepsilon}\sqrt{\frac{\beta^\star \log(KN)}{N}}\right)^{1-1/k} \right);$

*(ii) CTL:* $\mathrm{SubOpt}_{\mathrm{CTL}}(\beta^\star, k, \varepsilon, \alpha, N) \leq O\left( \alpha^{1-1/k} + \left(\frac{1}{\varepsilon}\sqrt{\frac{\beta^\star \log(KN)}{N}}\right)^{1-1/k} \right).$

For the case of $\alpha = 0$, as before one can simply choose to use the mean estimate result for $\alpha = 0$ as shown in Proposition 2 and adjust the burn-in period accordingly. This will lead to a bound that only has the second term in the above upper bounds. For $\beta^\star \geq 2$, one can observe that the upper bound of Algorithm 3 almost matches the lower bounds in Proposition 5 for both LTC and CTL settings. However, when $\beta^\star \in [1, 2)$ (i.e., good coverage case), it is known that the performance of LCB is worse than imitation learning, i.e., simply returning the most frequently selected arm in the offline dataset (when there is no privacy and corruption) (Rashidinejad et al., 2021). We leave it to future work to give a tight characterization of the sub-optimality when $\beta^\star \in [1, 2)$. Moreover, the proof of Proposition 6 also naturally gives us high-probability bounds without specifying $\delta = 1/N$ in the end.

## 6 Simulations

Beyond our theoretical results, we have also conducted a set of simulations for our three problems. Our theoretical results capture the worst-case performance (i.e., minimax rates). Thus, for simulations, we are particularly interested in the following two questions: (i) *Can we simulate the worst-case scenario and test the performance of our proposed algorithms?* and (ii) *How about their performance in non-worst-case scenarios?* We give detailed answers to both questions for all three problems in Appendix A, which offers additional insights into the interplay between privacy and robustness.

## 7 Concluding Remarks

To conclude, we have demonstrated an interesting interplay between privacy and robustness in three problems: mean estimation, online and offline MABs. The punchline across three problems is that

corruption after any LDP mechanism becomes easier, i.e., the same amount of corruption leads to a worse performance when compared to the case where Huber corruption happens before LDP mechanisms. We also give the first set of results for the most practical C-LDP-C setting.

Some interesting future directions include (i) improving the sub-optimal result for linear bandit in Charisopoulos et al. (2023) by following existing private linear bandits (Li et al., 2022a, 2024) along with the assumption of bounded reward; (ii) generalizing it to other privacy models such as shuffle DP (Chowdhury & Zhou, 2022c); (iii) studying the case where the heavy-tailedness is characterized by the central moment rather than the raw moment currently considered in our paper; (iv) extending the results to locally private and robust reinforcement learning by building upon existing results such as Chowdhury & Zhou (2022a); Liao et al. (2023); Zhou (2022).

## Acknowledgements

XZ is supported in part by NSF CNS-2153220 and CNS2312835. XZ would like to thank Daniel Kane for the helpful discussions.

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

# A  Simulations

In this section, we conduct numerical simulations to assess the performance of our algorithms in three problems (i.e., mean estimation, online MABs and offline MABs), under both LTC and CTL settings. Recall that our performance metrics for all three problems are minimax ones, which capture the worst-case performance. As a result, we are particularly interested in the following two questions in our simulations:

(i)  *Can we simulate the worst-case scenario and test the performance of our proposed algorithms?*

(ii)  *How about their performance in non-worst-case scenarios?*

Note that (i) essentially sheds further light on how to design the most powerful adversary Huber corruption model, which in turn could explain the separation result between LTC and CTL from the perspective of attacking. On the other hand, (ii) would help to illustrate our algorithms' performance in some mild/real-world non-adversary Huber corruption. For example, although the minimax regret for online MABs has a linear term in the worst case, the actual performance under the non-adversary corruption model can be sub-linear as we will show later.

## A.1  Mean estimation

We start with the worst-case scenario for the mean estimation under a large sample size regime where the minimax error rate is dominated by the corruption part, i.e., the separation result $(\alpha/\varepsilon)^{1-1/k}$ under LTC vs. $\alpha^{1-1/k}$ under CTL. To this end, we need to design the most powerful adversary corruption for both LTC and CTL. Here, we allow the (white-box) adversary to choose inlier distribution over $X$ and can adaptively choose Huber corruption distribution based on inlier distribution and the knowledge of our algorithm, e.g., LDP mechanism $Q$ in the LTC setting.

In particular, the adversary chooses the following inlier distribution:

$$P(X = 1/\gamma) = \frac{1}{2}\gamma^k, \quad P(X = -1/\gamma) = \frac{1}{2}\gamma^k, \quad P(X = 0) = 1 - \gamma^k \tag{5}$$

where $\gamma = (\alpha/\varepsilon)^{1/k}$ under LTC and $\gamma = (\alpha)^{1/k}$ under CTL. One can clearly see that $\mathbb{E}\left[|X|^k\right] \leq 1$ for all $k > 1$, hence $P \in \mathcal{P}_k$ for any $k > 1$ and $\alpha \leq \varepsilon$. Moreover, we have $\mathbb{E}[X] = 0$.

Now, we first consider the following strong Huber corruption model.

**Definition 7** (Strong Huber corruption for mean estimation). Let the inlier distribution over $X$ be given by (5). Under LTC: for each input $Y_i$, with probability $\alpha$, replace it with $M \cdot \frac{e^\varepsilon + 1}{e^\varepsilon - 1}$; Under CTL: for each input $X_i$, with probability $\alpha$, replace it with $M$;

Note that, the white-box adversary knows our algorithm and hence $M$. We are going to show that no matter how large the sample size is, the mean error has to be large for both LTC and CTL under the above strong Huber corruption.

Let us start with CTL and consider the sample size $n$ to be large. Then, according to Algorithm 1, $M = (1/\alpha)^{1/k} = 1/\gamma$, which leads to the fact that the mean of $Y$ is now $\alpha M = \alpha^{1-1/k}$ (note $\mathbb{E}[X] = 0$). Then, our estimator will essentially at best return the mean of $Y$, hence leading to the error of $\Omega(\alpha^{1-1/k})$. For LTC, with the choice of $\gamma$ and $M$, we also have $M = 1/\gamma$. By our design of LDP mechanism $Q$ in Algorithm 1, the mean of $Y$ is still zero and hence after the corruption, the mean of $Z$ becomes $\alpha \cdot M \frac{e^\varepsilon + 1}{e^\varepsilon - 1}$, which is the best outcome of our estimator, hence the error of $\Omega((\alpha/\varepsilon)^{1-1/k})$. Note that in both cases, the choice of corruption distribution needs care (i.e., adaptation to our algorithm), since otherwise, our estimator may still have an accurate estimate, as some other outlier values can be simply filtered out by our algorithm. More importantly, an alternative explanation of our separation result becomes evident: under LTC, the error is larger because the adversary has the capability to select a corruption value that is magnified by a factor of $1/\varepsilon$.

In our experiments, we choose $k = 2$ and consider various corruption level $\alpha \in \{0, 0.02, 0.05\}$ and privacy budget $\varepsilon \in \{0.3, 0.5, 1\}$. For each set of parameters, we conduct 300 runs and plot the average of the estimation error and corresponding confidence region. Fig. 2 illustrates our simulation results under strong Huber corruption in Definition 7. A common pattern behind all the plots in Fig. 2 is that due to strong corruption, the estimation error will only converge to a plateau and almost

match the lower bounds. Specifically, from the two plots in column (a), we see that when $\alpha = 0$ or $\varepsilon = 1$, the performance under LTC and CTL is close (i.e., no-separation), which aligns with our theoretical results. In the two plots of column (b), we see that LTC has a larger error than CTL and as $\varepsilon$ decreases (i.e., stronger privacy), the difference becomes larger, which matches our theoretical separation results. Finally, comparing the plots in column (c) with those in (b), we see that as the corruption level increases, the performance becomes worse.

We also consider the following weak corruption model, which simply flips the sign of the data.

**Definition 8** (Weak Huber corruption for mean estimation). Let the inlier distribution over $X$ be given by (5). Under LTC: for each input $Y_i$, with probability $\alpha$, replace it with $-Y_i$; Under CTL: for each input $X_i$, with probability $\alpha$, replace it with $-X_i$;

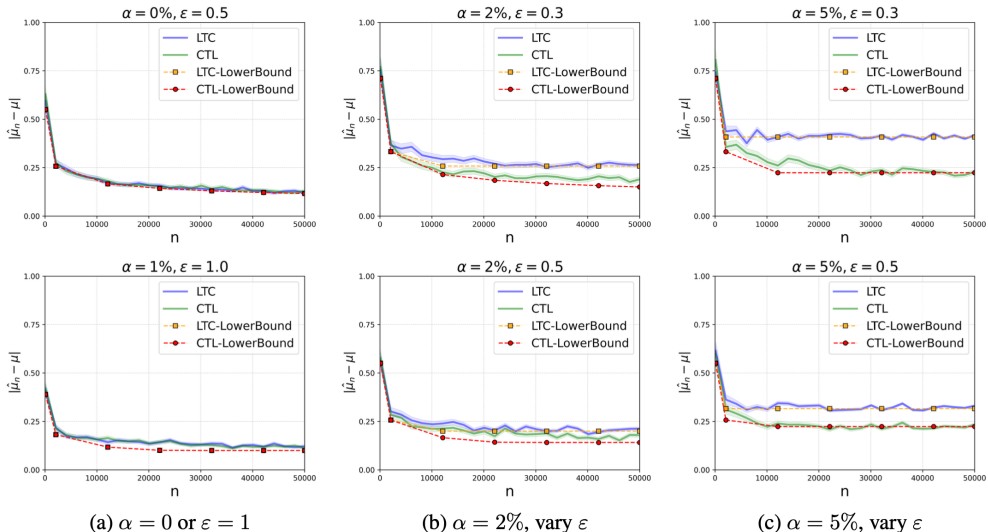

(a) $\alpha = 0$ or $\varepsilon = 1$          (b) $\alpha = 2\%$, vary $\varepsilon$          (c) $\alpha = 5\%$, vary $\varepsilon$

Figure 2: Mean estimation error with strong Huber corruption in Definition 7 under LTC and CTL settings.

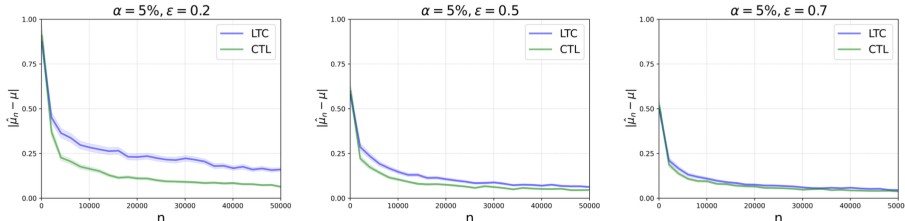

Figure 3: Mean estimation error with weak Huber corruption in Definition 8 under LTC and CTL settings

In Fig.3, we can see that under weak Huber corruption, the estimation error under our estimators can indeed decrease as the sample size increases. This demonstrates that in some real-world mild corruption scenarios, our estimators can yield promising performance.

## A.2    Online MABs

### A.2.1    Non-adversary Corruption

In this section, we first consider some classic heavy-tailed distributions under non-adversary corruption. The main purpose is to show that our proposed algorithm (i.e., Algorithm 2) can indeed achieve sublinear regret under certain scenarios. Moreover, our simulations will also provide some insights into our proof.

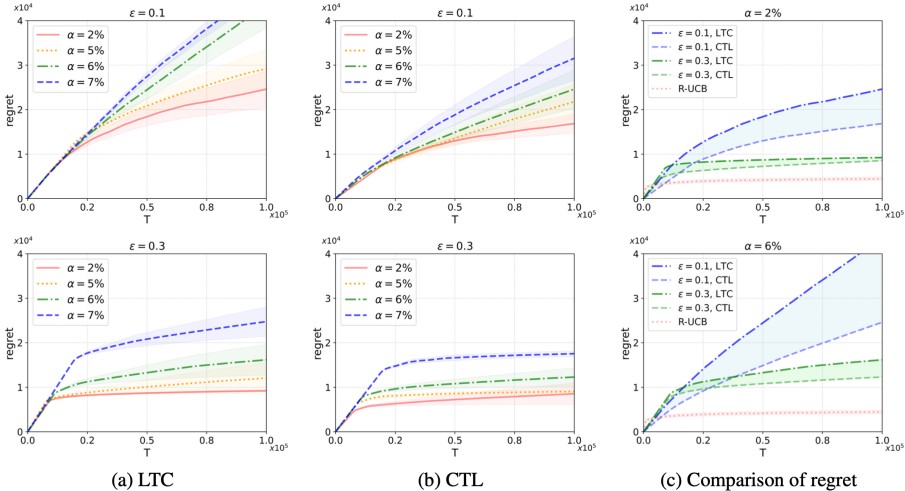

|   (a) LTC   |   (b) CTL   |   (c) Comparison of regret   |

Figure 4: Regret performance with weak Huber corruption in Definition 9 under LTC and CTL settings.

**Settings.** As in previous works Tao et al. (2022); Wu et al. (2023), we consider Pareto distribution, whose probability distribution is given by

$$f(x; x_m, s) = \begin{cases} \frac{s x_m^s}{x^{s+1}}, & \text{if } x \geq x_m \\ 0, & \text{otherwise} \end{cases}$$

where $s > 0$ is the shape parameter and $x_m > 0$ is the scale parameter. In our experiments, we consider there are $K = 10$ arms, and for each arm $i \in [K]$, the distribution is Pareto with $x_m = i$ and $s = 11$. To ensure that each arm's reward distribution is in $\mathcal{P}_k$ (i.e., $\mathbb{E}_{X \sim P}[|X|^k] \leq 1$), we normalize the reward by the $k$-th moment, which is $\frac{s x_m^k}{s-k}$. Consequently, the mean of each arm is $\frac{s-k}{x_m^{k-1}(s-1)}$. We consider $k = 2$, which along with our choices of $s$ and $x_m$, yields that arm 1 is the best arm with a mean of 0.9 while arm 10 is the worst arm with a mean of 0.09. For the corruption, we consider the following Huber model.

**Definition 9** (Huber corruption for online/offline MABs). Let each arm's inlier distribution be Pareto with the parameters described above. Under LTC, for each private view of reward from each $a \in [K]$, with probability $\alpha$, replace it with $M \cdot \frac{e^\varepsilon + 1}{e^\varepsilon - 1}$. Under CTL, for each raw reward from each arm $a \in [K]$, with probability $\alpha$, replace it with $M$.

*Remark* 6. It is worth noting that even though the above corruption values are the same as in Definition 7, it is not necessarily the worst-case as the inliers are now Pareto. That is, even after corruption, the agent can possibly still distinguish between different arms. We also consider strong corruption cases where after corruption, the agent cannot distinguish the distributions of two arms, hence a linear regret, see Fig. 6 for details.

Fig. 4 illustrates the regret performance of our proposed algorithm (i.e., Algorithm 2) for online MABs under LTC and CTL settings, with the specific corruption given by Definition 9. The two plots in column (a) capture the LTC setting while the two plots in column (b) denote the CTL setting. In both settings, we can see that for small corruption level $\alpha$, our algorithm can achieve sublinear regret, even though in the *worst-case* our minimax bounds are linear. In column (c), we also directly compare the regret performance under LTC and CTL with different sets of parameters of $\alpha$ and $\varepsilon$. As expected, the regret performance under LTC is worse than that under CTL, and as $\alpha$ increases or $\varepsilon$ decreases, the gap becomes larger. This demonstrates separation results in terms of actual performance rather than only in terms of theoretical upper bounds. As a baseline, we also compare with one classic robust MAB algorithm under heavy-tailed rewards proposed in Bubeck et al. (2013).

Fig. 5 compares our specific algorithms with the algorithm proposed in Tao et al. (2022), namely LDPRSE, which is proposed for the setting of LDP and heavy-tailed rewards in online MABs. Hence, our comparisons were made in the online MAB setting under weak corruption. The findings are organized into two columns, demonstrating the impact of varying $\alpha$ (corruption) values on

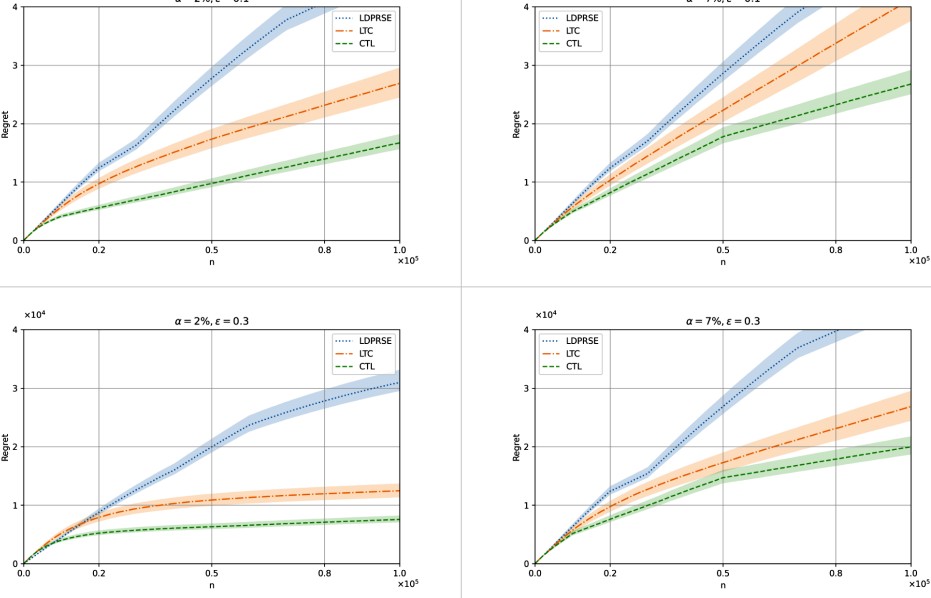

Figure 5: Comparison of Our Algorithms vs. LDPRSE in the online MAB setting under weak corruption.

performance as $\varepsilon$ (privacy) increases. These results highlight the advantages of our algorithms over LDPRSE in situations where there exist additional corruptions.

Note that our purpose in this section is not to demonstrate the superior performance of our proposed algorithm over all existing robust or/and private algorithms (given a large number of different existing ones). Rather, one of the goals is to use simulations to highlight the separation between LTC and CTL. Another important goal is to provide more insights into our proof of the regret upper bounds. Specifically, in our proof of the LTC setting (similar in CTL setting), we will divide the set of all sub-optimal arms $\mathcal{G}$ into two groups $\mathcal{G}_1$ and $\mathcal{G}_2$ where $\mathcal{G}_2 = \{a \in [K] \setminus a^* : c \left( \frac{\alpha}{\varepsilon} \right)^{1-1/k} \geq \frac{1}{2} \Delta_a \}$ for some constant $c$. Then, we argue that if $\mathcal{G}_2$ is empty, then one can still derive the standard logarithmic problem-dependent regret bound. This can also be somehow validated partially by our simulation results. In particular, under our problem instances described above, when $\alpha = 0.02$, $\varepsilon = 0.1$, and $c = 0.5$, we have $|\mathcal{G}_2| = 0$ under LTC (i.e., no sub-optimal arms in $\mathcal{G}_2$). In this case, as illustrated in the top plot of column (a) in Fig. 4, we can observe logarithmic order regret. This naturally extends to the larger $\varepsilon$ case, as illustrated in the bottom plot in column (a).

### A.2.2 Strong Huber Corruption

As mentioned above, we also create a strong Huber corruption for online MABs, in this case, the regret becomes linear which matches our minimax lower bound. In this scenario, our goal is to create an adversary strong Huber corruption for online MABs, where the agent cannot distinguish the distributions of two arms by utilizing the following probability distribution:

$$P(X = 1/\gamma) = \gamma^k, \quad P(X = 0) = 1 - \gamma^k$$
$$P'(X = 1/\gamma) = \gamma^k/2, \quad P'(X = 0) = 1 - \gamma^k/2$$

where $\gamma$ adopts the form $c_1 \cdot (\alpha/\varepsilon)^{1/k}$ under LTC and $c_1 \cdot (\alpha)^{1/k}$ under CTL, with $c_1$ configured as 0.1 to ensure $\gamma^k \leq 1$ for an expansive $\alpha$. As before, $P, P' \in \mathcal{P}_k$ for any $k > 1$ and $\mu(P) = \gamma^{k-1}$, $\mu(P) = \gamma^{k-1}/2$. Let $P$ and $P'$ represent the distributions for arms 0 and 1 respectively. We define the corruption distribution under CTL settings as:

**Definition 10** (Strong Huber corruption under CTL Settings).

$$C(X = 1/\gamma) = \gamma^k/2, \qquad C(X = 0) = 1 - \gamma^k/2$$
$$C'(X = 1/\gamma) = \gamma^k/(2\alpha), \qquad C'(X = 0) = 1 - \gamma^k/(2\alpha)$$

According to 2, it is apparent that the agent cannot differentiate between $P$ and $P'$ upon executing the operation:

$$(1 - \alpha)P + \alpha C = (1 - \alpha)P' + \alpha C'$$

This outcome emerges from the CTL's inherent nature of initially introducing contamination, which perseveres in maintaining indistinguishability, even post-transmission through the LDP channel and the Huber model.

In the context of LTC settings, the distinctiveness arises from the fact that the distributions of $P$ and $P'$ undergo alterations after passing through LDP, necessitating corresponding corruptions. Let $R$ and $R'$ be the post-LDP transformation distributions over variable Y, defined as:

$$R(Y = S) = \frac{1}{2} + \frac{\gamma^k}{2} \cdot \frac{e^\varepsilon - 1}{e^\varepsilon + 1}, \quad R(Y = -S) = \frac{1}{2} - \frac{\gamma^k}{2} \cdot \frac{e^\varepsilon - 1}{e^\varepsilon + 1}$$

$$R'(Y = S) = \frac{1}{2} + \frac{\gamma^k}{4} \cdot \frac{e^\varepsilon - 1}{e^\varepsilon + 1}, \quad R'(Y = -S) = \frac{1}{2} - \frac{\gamma^k}{4} \cdot \frac{e^\varepsilon - 1}{e^\varepsilon + 1}$$

where $S = M \cdot \frac{e^\varepsilon + 1}{e^\varepsilon - 1}$.

Additionally, we define the corruption distribution as:

**Definition 11** (Strong Huber corruption under LTC Settings).

$$N(Y = S) = \frac{\gamma^k}{4} \cdot \frac{e^\varepsilon - 1}{e^\varepsilon + 1}, \quad N(Y = -S) = 1 - \frac{\gamma^k}{4} \cdot \frac{e^\varepsilon - 1}{e^\varepsilon + 1}$$

$$N'(Y = S) = \frac{\gamma^k}{4} \cdot \frac{e^\varepsilon - 1}{e^\varepsilon + 1} \cdot \frac{1}{\alpha}, \quad N'(Y = -S) = 1 - \frac{\gamma^k}{4} \cdot \frac{e^\varepsilon - 1}{e^\varepsilon + 1} \cdot \frac{1}{\alpha}$$

Now we also have $(1 - \alpha)R + \alpha N = (1 - \alpha)R' + \alpha N'$, indicating our continued inability to distinguish between $P$ and $P'$ in the LTC setting.

Fig. 6 illustrates the regret performance of our proposed algorithm (i.e., Algorithm 2) for online MABs under LTC and CTL settings, with the strong Huber corruption in Definition 10 and Definition 11. A common pattern behind all the plots in Fig. 6 is that due to strong huber corruption, the agent cannot distinguish the distributions of two arms, hence linear regret. Based on the analysis above, we anticipate that the regret will scale linearly by a factor of $c_1$ with respect to our minimax clean regret and Fig. 6 aligned with our discussion. The two plots in column (a) capture the LTC setting while the two plots in column (b) denote the CTL setting. As expected, the regret performance under LTC is worse than that under CTL, highlighting separation results in terms of actual performance rather than only in terms of theoretical upper bounds.

## A.3 Offline MABs

In the offline case, the analyzer/agent is given a batch of pre-collected data with private and corrupted view. In our experiments, we again consider the case that there are $K = 10$ arms and each arm's raw reward distribution is Pareto with the same parameters as in the online case. For corruption, we again consider the one given by Definition 9.

One difference here is that we need to specify the behavior policy $\pi$ that is used to collect the data. To this end, we consider the following policy $\pi$ in our simulation results: for each sample size $N$, we pulled the best arm (i.e., arm 1) $\frac{N}{3}$ times and each other arm $i \neq 1$ uinformly, i.e., $\frac{2N}{3(K-1)}$ times. That is, roughly speaking, we approximately have $1/\pi(a^\star) = 3$, which aligns with our theoretical assumption (i.e., the finite concentrability coefficient $\beta^\star \geq 2$ when our upper bounds are tight in minimax sense).

Fig. 7 illustrates the suboptimality of our algorithm (i.e., Algorithm 3) under both LTC and CTL settings. We can see that in both settings, the sub-optimality could approach zero under several values of privacy parameters. This again highlights that under mild/non-adversary corruption, the algorithm could yield reasonably good performance, rather than the pessimistic worst-case one. Also, we observe that even in this non-adversary corruption case, suboptimality under LTC in general is still worse than that under CTL. Finally, it is not surprising that for both LTC and CTL, as $\alpha$ increases or $\varepsilon$ decreases, sub-optimality will increase.

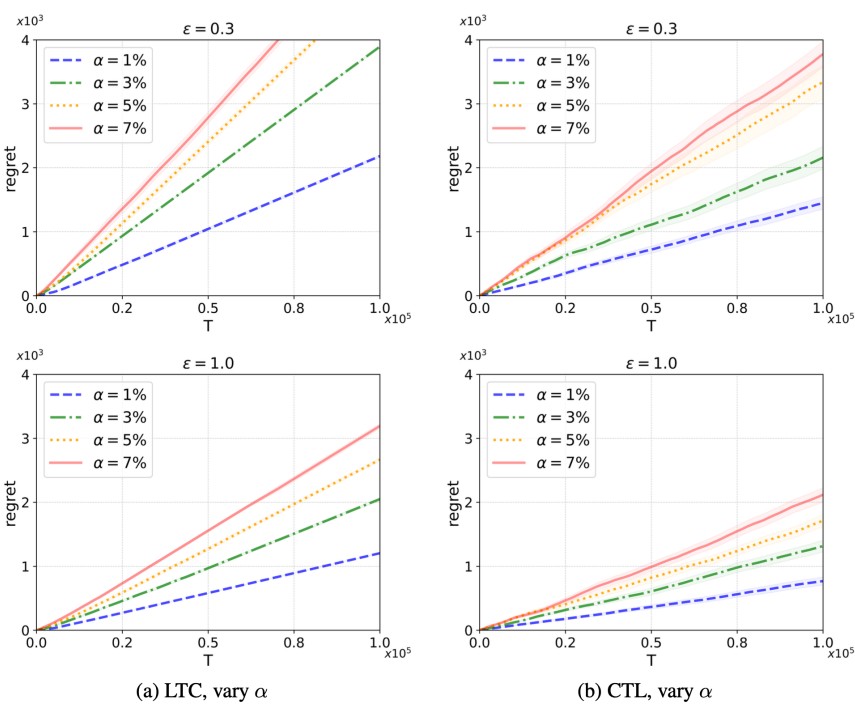

(a) LTC, vary $\alpha$          (b) CTL, vary $\alpha$

Figure 6: Regret performance with strong Huber corruption in Definition 10 unde CTL settings and Definition 11 under LTC settings.

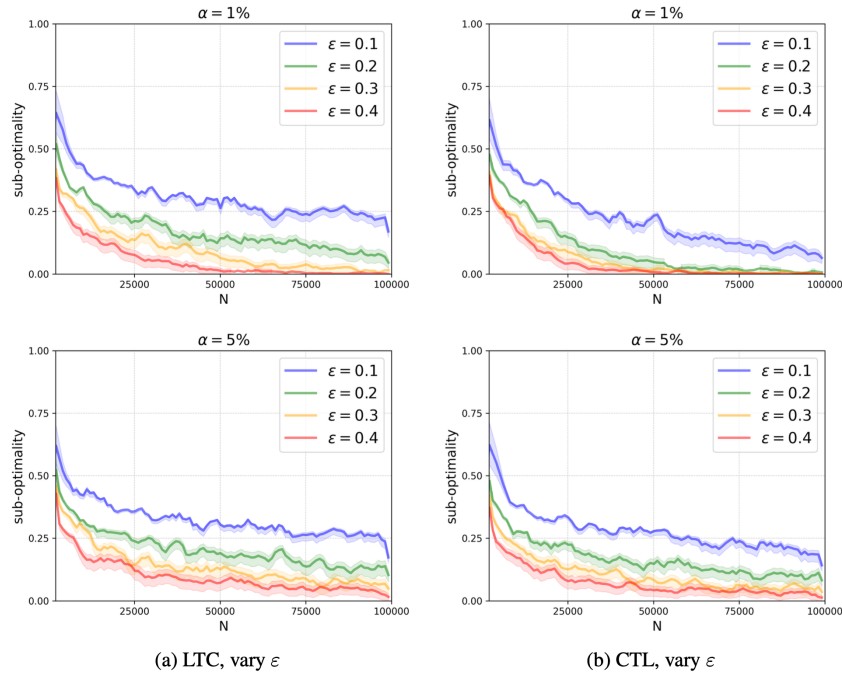

(a) LTC, vary $\varepsilon$          (b) CTL, vary $\varepsilon$

Figure 7: Suboptimality performance with Huber corruption in Definition 9 under LTC and CTL settings.

# B Additional Related Work

**Private MABs.** To offer mathematically rigorous privacy protection, LDP is first introduced to MABs in Ren et al. (2020) where the authors establish private lower bounds on both problem-dependent and problem-independent (minimax) regrets as well as several LDP mechanisms and learning algorithms that achieve nearly-optimal performance. Later, it is generalized to the heavy-tailed setting in Tao et al. (2022). LDP has also been considered in various other bandit settings (Chen et al., 2020; Zheng et al., 2020; Zhou & Tan, 2021). In addition to LDP, other strictly weaker privacy models have also been considered in MABs to achieve a better regret, such as central DP where users need to trust the central learner (Mishra & Thakurta, 2015; Tossou & Dimitrakakis, 2016; Sajed & Sheffet, 2019) and distributed DP where users need to trust the intermediate third-party (Tenenbaum et al., 2021; Chowdhury & Zhou, 2022b). In addition to the above online MABs, recent work (Qiao & Wang, 2022) also considers offline RL (hence MABs) under central DP with bounded rewards.

**Robust MABs.** Robust MABs under Huber corruption have been recently studied in Kapoor et al. (2019); Mukherjee et al. (2021); Basu et al. (2022); Agrawal et al. (2024). Several other corruption models have also been considered in MABs, such as budgeted-corruption model where the cumulative difference between observed reward and true reward is bounded by some constant budget (Lykouris et al., 2018; Gupta et al., 2019) and strong contamination model (Niss & Tewari, 2020; Altschuler et al., 2019). Robust regret minimization in MABs under heavy-tailed rewards have also been studied, e.g., Bubeck et al. (2013); Agrawal et al. (2021).

**Private and Robust MABs.** As mentioned above, the existing literature largely investigate privacy and robustness in MABs separately. To the best of our knowledge, there are only two very recent works that consider privacy and robustness in MABs simultaneously. In Wu et al. (2023), the authors consider the central DP model where the raw non-private feedback received by the central learner can be first corrupted under Huber model. This is in sharp contrast to our local DP model, which is not only stronger but allows us to study the order of corruption and privacy. In Charisopoulos et al. (2023), the authors study linear bandits (which includes MAB as a special case) under LDP and then Huber corruption (i.e., LTC setting). As discussed in Section 4, their regret bound is sub-optimal and worse than ours when reduced to the MAB case. Note that we also study the CTL setting, which in turn highlights the interplay between privacy and corruption. Moreover, the results for both LTC and CTL allow us to give the first results for the C-LDP-C setting.

**Private and Robust Mean Estimation.** Our work is inspired by recent advances in (locally) private and robust mean estimation. In particular, for the CTL setting, the authors of Li et al. (2022b) give the tight characterization in terms of mean-square-error (MSE). In contrast, we derive the high probability concentration. For LTC, both Cheu et al. (2021); Chhor & Sentenac (2023) give constant-probability concentration when the inlier distribution is bounded. Instead, we present the high-probability version even for heavy-tailed inlier distribution, which requires new analysis and design of the estimators. We would also like to point out some other related private and/or robust mean estimation results. For instance, under central DP, Kamath et al. (2020) gives the first high probability mean concentration for heavy-tailed distributions. For standard non-private mean estimation under heavy tails, we refer readers to the nice survey by Lugosi & Mendelson (2019). For non-private mean estimation under corruption in general high-dimension space, we refer readers to the nice book by Diakonikolas & Kane (2023). We finally remark that there are recent exciting advances in understanding the connection between robustness and privacy in mean estimation (e.g., robustness induces privacy Hopkins et al. (2023); Asi et al. (2023) and vice versa Georgiev & Hopkins (2022)), which, however, mainly focus on the central DP model.

# C Discussions

## C.1 Discussions on Practical Scenarios for LTC and CTL

In the introduction, we have motivated our paper using the example of online recommendation/advertising via MABs. Here we give two more concrete examples. The key difference between LTC and CTL in practice is that LTC mainly models the situation where the data transmission is vulnerable to manipulation while CTL models the situation where the data source is more vulnerable to manipulation.

**CTL:** Consider a healthcare recommendation system that suggests personalized health interventions based on patient data. In this case, the data might first be corrupted (intentionally or unintentionally) before being subjected to LDP mechanisms, such as when data is collected from various sources with different levels of reliability or when users self-report their health information with errors or falsifications. However, the data transmission is often well-controlled in this case and is not likely vulnerable to manipulation due to strong federal regulations.

**LTC:** Consider a wireless IoT (Internet-of-Things) smart-home application where sensors are deployed to monitor/control the temperature or other metrics in homes. These sensors often have built-in checks to ensure that the data is collected correctly. However, after the LDP mechanism at each sensor from each home, the data transmission process through wireless networks (channels) is often more vulnerable to manipulation attacks, e.g., man-in-the-middle attacks, packet sniffing, or spoofing.

In addition to the above two examples, we do believe that there are many other practical scenarios that motivate our study of the interplay between LDP and Huber corruption.

**Key implication of "LTC is harder":** If our recommendation system requires LDP protection, then the adversary can tailor its manipulation attack (corruption) based on the LDP mechanism (hence $\varepsilon$) to amplify the error by the order of $1/\varepsilon$. In other words, LDP protocols are highly vulnerable to manipulation – poisoning the private messages can be far more destructive than poisoning the data itself. As a result, it is important to keep our private protocol "secret" as the adversary needs to tailor its attack according to the LDP protocol to create the worst-case scenario (strongest attack).

### C.2 Robust Estimators without Knowing Corruption Parameter

Currently, whether it is possible to derive a tight error bound without knowing $\alpha$ is still unclear to us. In particular, on the one hand, there are some positive results (Jain et al., 2022; Bhatt et al., 2022) for some estimators. On the other hand, some work suggests some negative results regarding MAB problems (Agrawal et al., 2024). Note that all Jain et al. (2022); Bhatt et al. (2022); Agrawal et al. (2024) only consider corruption, i.e., no privacy protection. Thus, one interesting future work is to settle down this problem, which is beyond the scope of our current paper.

### C.3 Ungrounded Regret Upper Bound in State-of-the-Art

In Tao et al. (2022), the authors consider a simpler setting – locally private heavy-tailed online MABs, i.e., without corruption. They claimed to achieve a regret upper bound on the order of $O\left(\left(\frac{K}{\varepsilon^2}\right)^{1-1/k} T^{1/k}\right)$. However, given our tighter lower bound $\Omega\left(T^{\frac{k+1}{2k}}\left(\frac{K}{\varepsilon^2}\right)^{\frac{k-1}{2k}}\right)$ in Proposition 3, their upper bound becomes ungrounded as it contradicts our lower bound for large $T$. In particular, considering $k = 2$ (with only a bounded second moment), our lower bound gives a regret on the order of $\Omega(T^{3/4})$ while their upper bound is $O(\sqrt{T})$.

*Remark* 7. In fact, our lower bound also gives another interesting interplay between privacy and robustness (in particular, heavy-tailed rewards). Specifically, in the non-private case, as shown in the Bubeck et al. (2013), one can still achieve $\Theta(\sqrt{T})$ regret when the reward distributions have only bounded second moments. However, in the locally private case, our lower bound indicates that the regret is at least $\Omega(T^{3/4})$.

## D  Broader Impact Statement

This research presents novel insights into the interplay between local differential privacy and robustness in the context of Multi-Armed Bandits (MABs), with a focus on two distinct settings: Local Differential Privacy then Corruption (LTC) and Corruption then Local Differential Privacy (CTL). The findings have broad implications in various domains, particularly in online advertising and recommendation systems, where privacy preservation and data integrity are paramount. By enhancing the robustness of MAB algorithms against corruption and heavy-tailed feedback while ensuring local privacy, our work can significantly contribute to the development of more secure and reliable decision-making systems. We show that the mean estimation error under LTC is larger than under CTL, emphasizing that LTC is a more challenging setting. This separation is critical for

practical applications like healthcare recommendation systems (CTL) and wireless IoT smart-home applications (LTC). Additionally, our algorithms can adaptively guarantee optimal minimax rates across different settings without prior knowledge, which is crucial for real-world scenarios where the specific setting may not be known in advance. However, the complexity and computational demands of these advanced algorithms might limit their accessibility to smaller organizations, potentially widening the gap between large and small entities. Moreover, while our approach reduces privacy leakage and data manipulation risks, it does not completely eliminate them. This is particularly important because adversaries can tailor their attacks based on the LDP mechanism to amplify errors. Thus, ongoing efforts should focus on further improving these algorithms to address potential ethical issues, including data bias and privacy concerns, and enhancing their accessibility and fairness. Furthermore, deriving tight error bounds without knowing the corruption parameter $\alpha$ remains an open challenge, suggesting the need for future research in this area.

## E   Proof of Proposition 1

*Proof.* We first focus on the LTC setting and divide the proof into two steps.

**Step 1: Without corruption.** By definition, it suffices to establish a lower bound on the concentration even without corruption. That is, under LTC, $Z_i = Y_i$ for all $i \in [n]$. This will give us the second term in the bound.

Consider the following two distributions $P$ and $P'$. Let $\gamma > 0$, specified later and

$$
\begin{aligned}
P(X = 1/\gamma) &= \gamma^k, \quad P(X = 0) = 1 - \gamma^k \\
P'(X = 1/\gamma) &= 1/2 \cdot \gamma^k, \quad P'(X = 0) = 1 - 1/2 \cdot \gamma^k.
\end{aligned}
\tag{6}
$$

It is easy to see that for both $P, P'$, $\mathbb{E}\left[|X|^k\right] \leq 1$ for all $k > 1$, hence $P, P' \in \mathcal{P}_k$ for any $k > 1$. Moreover, we have $|\mu(P) - \mu(P')| = 1/2 \cdot \gamma^{k-1}$ and $\mathrm{TV}(P, P') = 1/2 \cdot \gamma^k$. For any $\varepsilon$-LDP channel $Q$, let $M$ and $M'$ be the induced marginal distribution from $P$ and $P'$, respectively. That is, for $i \in [n]$, $Y_i \sim M$ and $Y_i' \sim M'$. Let $Y_{[n]} = \{Y_i\}_{i=1}^n$ and $Y_{[n]}' = \{Y_i'\}_{i=1}^n$, i.e., $Y_{[n]} \sim M^{\otimes n}$ and $Y_{[n]}' \sim M'^{\otimes n}$.

The high-level idea behind our proof is as follows: Given any sample size $n$, if there exists at least probability $2\delta$ such that $Y_{[n]} = Y_{[n]}'$, then one has to incur $\Omega(\gamma^{k-1})$ estimation error with probability $\delta$. This naturally reminds us to think about maximal coupling, since it maximizes the probability that $Y_{[n]} = Y_{[n]}'$ and is also closely related to TV distance. In particular, we have the following textbook facts.

**Lemma 1.** *Let $P_1$ and $P_2$ be two distributions on $\mathcal{X}$ that share the same $\sigma$-algebra. There exists a coupling $\omega^*(P_1, P_2)$, which is a distribution over $\mathcal{X}^2$ such that*

$$
\begin{aligned}
\mathbb{P}_{(X_1, X_2) \sim \omega^*(P_1, P_2))}(X_1 \neq X_2) &= \mathrm{TV}(P_1, P_2) \\
\forall S \text{ measurable}, \mathbb{P}_{(X_1, X_2) \sim \omega^*(P_1, P_2))}(X_1 \in S) &= P_1(X_1 \in S) \\
\forall S \text{ measurable}, \mathbb{P}_{(X_1, X_2) \sim \omega^*(P_1, P_2))}(X_2 \in S) &= P_2(X_2 \in S).
\end{aligned}
$$

*This coupling is called* maximal coupling.

Based on this fact, fix some $n$, if $(Y_{[n]}, Y_{[n]}')$ is sampled from the maximal coupling $\omega^*(M^{\otimes n}, M'^{\otimes n})$, then we know that there exists a probability $p = 1 - \mathrm{TV}(M^{\otimes n}, M'^{\otimes n})$ such that $Y_{[n]} = Y_{[n]}'$. To lower bound $p$, we need to upper bound the TV distance. To this end, we will leverage Bretagnolle–Huber inequality and strong data processing inequality (i.e., Corollary 3 in Duchi et al. (2018)). In particular,

we have

$$\text{TV}(M^{\otimes n}, M'^{\otimes n}) \overset{(a)}{\leq} 1 - \frac{1}{2}\exp\left(-\text{KL}\left(M^{\otimes n} \| M'^{\otimes n}\right)\right)$$

$$\overset{(b)}{=} 1 - \frac{1}{2}\exp\left(-4(e^{\varepsilon} - 1)^2 \cdot n \cdot (\text{TV}(P, P'))^2\right)$$

$$= 1 - \frac{1}{2}\exp\left(-4(e^{\varepsilon} - 1)^2 \cdot n \cdot \gamma^{2k}\right)$$

$$\overset{(c)}{\leq} 1 - \frac{1}{2}\exp\left(-16\varepsilon^2 \cdot n \cdot \gamma^{2k}\right),$$

where (a) holds by Bretagnolle–Huber inequality; (b) holds by Corollary 3 in Duchi et al. (2018); (c) is true since $e^{\varepsilon} - 1 \leq 2\varepsilon$ for $\varepsilon \in [0, 1]$. Thus, let $\gamma = c_1 \left(\frac{\sqrt{\log(1/\delta)}}{\varepsilon\sqrt{n}}\right)^{1/k}$ for some constant $c$. Then, for large enough $n$, $\gamma^{k-1} < 1$ and $\text{TV}(M^{\otimes n}, M'^{\otimes n}) \leq 1 - 2\delta$, which implies that with probability at least $\delta$, the error is $\Omega(\gamma^{k-1}) = \Omega\left(\left(\frac{\sqrt{\log(1/\delta)}}{\varepsilon\sqrt{n}}\right)^{1-1/k}\right)$.

**Step 2: Corruption part.** Recall that under $\alpha$-Huber, for each private view $Y_i$, it is independently corrupted with probability $\alpha$, and when it happens, $Z_i$ is sampled from an arbitrary noise distribution $N$; otherwise, $Z_i = Y_i$. To proceed, we will utilize the following useful fact.

**Lemma 2** (Theorem 5.1 in Chen et al. (2018)). *Let $R_1$ and $R_2$ be two distributions on $\mathcal{X}$; If for some $\alpha \in [0, 1/2)$, we have that $\text{TV}(R_1, R_2) \leq \frac{\alpha}{1-\alpha}$, then there exist two distributions $N_1$ and $N_2$ on the same probability space such that*

$$(1 - \alpha)R_1 + \alpha N_1 = (1 - \alpha)R_2 + \alpha N_2.$$

This result says that the Huber model with parameter $\alpha$ can corrupt two distributions that are close in TV distance so that the outputs are essentially sampled from the same distribution, hence indistinguishable.

Another fact we will leverage is that LDP mechanism is a "contraction" in that it will make the TV distance closer.

**Lemma 3** (Corollary 2.9 in Kairouz et al. (2014)). *For any $\varepsilon > 0$, let $Q$ be any $\varepsilon$-LDP mechanism. Then, for any pair of distributions $P_1$ and $P_2$, the induced marginals $M_1$ and $M_2$ satisfy*

$$\text{TV}(M_1, M_2) \leq \frac{e^{\varepsilon} - 1}{e^{\varepsilon} + 1}\text{TV}(P_1, P_2).$$

The above fact indicates that for $\varepsilon \in [0, 1]$, $\text{TV}(M_1, M_2) \leq O(\varepsilon)\text{TV}(P_1, P_2)$. With the above two facts, it suffices for us to find two distributions $P$ and $P'$ for $X_i$ with a "large" mean difference, such that the induced marginal distributions for $Y_i$ is $O(\alpha)$. To this end, we again consider the two distributions in (6) with a different choice of $\gamma$. Since $\text{TV}(P, P') = 1/2 \cdot \gamma^k$, by Lemma 3, choosing $\gamma = c' \cdot (\alpha/\varepsilon)^{1/k}$ for some small constant $c' > 0$ yields that $\text{TV}(M, M') \leq \alpha \leq \alpha/(1 - \alpha)$. Hence, by Lemma 2, there exists Huber contamination such that it is impossible to distinguish the final outputs. Hence, with a probability of at least $1/2$, the error is $\Omega(\gamma^{k-1}) = \Omega((\alpha/\varepsilon)^{1-1/k})$. We finally conclude that for any $\delta \in (0, 1/2)$, with probability at least $\delta$, for all large enough $n$, estimation error is $\Omega(\gamma^{k-1}) = \Omega((\alpha/\varepsilon)^{1-1/k})$. This finishes the proof for the LTC setting.

As for the CTL setting, the second term in the lower bound follows the same proof as in Step 1. The key difference lies in Step 2, i.e., the first term in the bound. In particular, since the contamination is before LDP, one can now only choose $\gamma = c'\alpha^{1/k}$, i.e., no "contraction" from LDP anymore. As a result, the estimation error is $\Omega(\gamma^{k-1}) = \Omega(\alpha^{1-1/k})$. $\qquad\square$

## F  Proof of Proposition 2

*Proof.* Let us start with the LTC setting. As for privacy, it builds on the privacy guarantee of random response.

**Privacy.** By definition, we need to show that for any two inputs $x, x' \in \mathcal{X}$ and $y \in \left\{ M \frac{e^\varepsilon + 1}{e^\varepsilon - 1}, -M \frac{e^\varepsilon + 1}{e^\varepsilon - 1} \right\}$

$$\frac{\mathbb{P}[Y = y | X = x]}{\mathbb{P}[Y = y | X = x']} \leq e^\varepsilon.$$

Consider the case $y = M \frac{e^\varepsilon + 1}{e^\varepsilon - 1}$ and similar analysis applies to the other case. Let $P_{x \to M^+}$ be the probability that $x$ is translated to $M$ in our mechanism $Q$ and $P_{x \to M^-}$ be the probability that $x$ is translated to $-M$ in our mechanism $Q$. Similarly defines $P_{x' \to M^+}$ and $P_{x' \to M^-}$.

Thus, according to our $Q$ in Algorithm 1) and let $P_\varepsilon := \frac{e^\varepsilon}{e^\varepsilon + 1}$, we have

$$\mathbb{P}[Y = y | X = x] = P_{x \to M^+} P_\varepsilon + P_{x \to M^-}(1 - P_\varepsilon)$$
$$\mathbb{P}[Y = y | X = x'] = P_{x' \to M^+} P_\varepsilon + P_{x' \to M^-}(1 - P_\varepsilon)$$

As a result,

$$\frac{\mathbb{P}[Y = y | X = x]}{\mathbb{P}[Y = y | X = x']} = \frac{P_{x \to M^+} P_\varepsilon + P_{x \to M^-}(1 - P_\varepsilon)}{P_{x' \to M^+} P_\varepsilon + P_{x' \to M^-}(1 - P_\varepsilon)} \leq \frac{P_\varepsilon}{1 - P_\varepsilon} \leq e^\varepsilon.$$

**Utility.** For the utility part, we will divide the proof into four steps.

We draw the following informal diagram for an illustration of Algorithm 1.

$$X_i \xrightarrow{\text{Trunc.}(M)} \bar{X}_i \xrightarrow{\text{Random Rounding}} X'_i \xrightarrow{\text{Random Response}} Y_i \xrightarrow{\text{Corruption}} Z_i \xrightarrow{\text{Trunc.}(M \frac{e^\varepsilon + 1}{e^\varepsilon - 1})} \bar{Z}_i \xrightarrow{\text{Sample Mean}} \widehat{\mu}_n$$

**Step 1:** Bound the number of corrupted points.

By Chernoff bound for the binomial distribution, we have that for $n \geq 3 \log(1/\delta)/\alpha$

$$|\mathcal{B}| \leq 2\alpha n, \quad w.p. \quad 1 - \delta,$$

where $|\mathcal{B}|$ denotes the total number of corrupted ("bad") points. Let this event be $\mathcal{E}$, and in the following steps, we will condition on this event.

**Step 2:** Bound the distance $|\mathbb{E}[X'_i] - \mathbb{E}[X_i]|$.

$$
\begin{aligned}
|\mathbb{E}[X_i] - \mathbb{E}[X'_i]| &\leq |\mathbb{E}[X_i] - \mathbb{E}[\bar{X}_i]| + |\mathbb{E}[\bar{X}_i] - \mathbb{E}[X'_i]| \\
&\overset{(a)}{=} |\mathbb{E}[X_i] - \mathbb{E}[\bar{X}_i]| + 0 \\
&\leq \mathbb{E}[|X_i| \mathbb{1}(|X_i| \geq M)] \\
&\overset{(b)}{\leq} \frac{1}{M^{k-1}}
\end{aligned}
$$

where (a) holds by the property of random rounding. Recall that, for any $\bar{X}_i \in [-M, M]$, $X'_i = M$ w.p. $\frac{1 + \bar{X}_i/M}{2}$ and $X'_i = -M$ w.p. $\frac{1 - \bar{X}_i/M}{2}$. Thus, one can see $\mathbb{E}[X'_i | \bar{X}_i] = \bar{X}_i$, hence $\mathbb{E}[\bar{X}_i] = \mathbb{E}[X'_i]$; (b) holds by Hölder's inequality and the fact $k$-th moment of $X_i$ is upper bounded by one.

**Step 3:** Bound the distance $|\mathbb{E}[X'_i] - \widehat{\mu}_n|$.

$$
\begin{aligned}
|\widehat{\mu}_n - \mathbb{E}[X'_i]| &= |\frac{1}{n} \sum_i \bar{Z}_i - \mathbb{E}[X'_i]| \\
&= |\frac{1}{n} \sum_i \bar{Z}_i - \frac{1}{n} \sum_i Y_i + \frac{1}{n} \sum_i Y_i - \mathbb{E}[X'_i]| \\
&\overset{(a)}{\leq} 2\alpha \cdot M \cdot \frac{e^\varepsilon + 1}{e^\varepsilon - 1} + |\frac{1}{n} \sum_i Y_i - \mathbb{E}[X'_i]| \\
&\overset{(b)}{\leq} 2\alpha \cdot M \cdot \frac{e^\varepsilon + 1}{e^\varepsilon - 1} + O\left( M \cdot \frac{e^\varepsilon + 1}{e^\varepsilon - 1} \cdot \sqrt{\frac{\log(1/\delta)}{n}} \right) \quad w.p. \quad 1 - \delta
\end{aligned}
$$

where (a) holds by triangle inequality, the event $\mathcal{E}$ in step 1, and the fact that $\bar{Z}_i$, $Y_i$ are both bounded; (b) holds by Hoeffding inequality. Note that $Y_i = \frac{e^\varepsilon + 1}{e^\varepsilon - 1} X_i'$ w.p. $\frac{e^\varepsilon}{e^\varepsilon + 1}$ and $Y_i = -\frac{e^\varepsilon + 1}{e^\varepsilon - 1} X_i'$ w.p. $\frac{1}{e^\varepsilon + 1}$. That is, $\mathbb{E}[Y_i] = \mathbb{E}[X_i']$ and $Y_i = \{M \cdot \frac{e^\varepsilon + 1}{e^\varepsilon - 1}, -M \cdot \frac{e^\varepsilon + 1}{e^\varepsilon - 1}\}$.

**Step 4:** Put the above two parts together.

For any $\varepsilon \in [0, 1]$, any $\delta \in (0, 1)$ and any $P \in \mathcal{P}_k$, we have with probability at least $1 - \delta$,

$$|\widehat{\mu}_n - \mu(P)| \leq O\left(\frac{1}{M^{k-1}} + \frac{\alpha M}{\varepsilon} + \frac{M}{\varepsilon}\sqrt{\frac{\log(1/\delta)}{n}}\right).$$

Thus, choosing $M = \min\left\{\left(\frac{\varepsilon}{\alpha}\right)^{1/k}, \left(\frac{\sqrt{n}\varepsilon}{\sqrt{\log(1/\delta)}}\right)^{1/k}\right\}$, yields that

$$|\widehat{\mu}_n - \mu(P)| \leq O\left(\left(\frac{\alpha}{\varepsilon}\right)^{1-1/k} + \left(\frac{1}{\varepsilon}\sqrt{\frac{\log(1/\delta)}{n}}\right)^{1-1/k}\right),$$

which finishes the proof for the LTC setting.

Now, let us move to the CTL setting. For privacy, it follows from the same idea as in the LTC setting.

For utility, we will divide the proof into five steps and leverage the following informal diagram for an illustration of Algorithm 1.

$$X_i \xrightarrow{\text{Corruption}} Y_i \xrightarrow{\text{Trunc.}(M)} \bar{Y}_i \xrightarrow{\text{Random Rounding}} Y_i' \xrightarrow{\text{Random Response}} Z_i \xrightarrow{\text{Sample Mean}} \widehat{\mu}_n$$

**Step 1:** Bound the number of corrupted points.

By Chernoff bound for the binomial distribution, we have that for $n \geq 3\log(1/\delta)/\alpha$

$$|\mathcal{B}| \leq 2\alpha n, \quad w.p. \quad 1 - \delta,$$

where $|\mathcal{B}|$ denotes the total number of corrupted ("bad") points. Let this event be $\mathcal{E}$, and in the following steps, we will condition on this event.

**Step 2:** Bound the distance $|\widehat{\mu}_n - \frac{1}{n}\sum_i \mathbb{E}[\bar{Y}_i]|$.

$$|\widehat{\mu}_n - \frac{1}{n}\sum_i \mathbb{E}[\bar{Y}_i]| \stackrel{(a)}{=} |\frac{1}{n}\sum_i Z_i - \frac{1}{n}\sum_i \mathbb{E}[Y_i']|$$
$$\stackrel{(b)}{\leq} O\left(M \cdot \frac{e^\varepsilon + 1}{e^\varepsilon - 1} \cdot \sqrt{\frac{\log(1/\delta)}{n}}\right) \quad w.p. \quad 1 - \delta,$$

where (a) holds by property of random rounding, i.e., $\mathbb{E}[\bar{Y}_i] = \mathbb{E}[Y_i']$; (b) holds by property of random response, i.e., $\mathbb{E}[Z_i] = \mathbb{E}[Y_i']$ and Hoeffding inequality.

**Step 3:** Bound the distance $|\frac{1}{n}\sum_i \bar{Y}_i - \frac{1}{n}\sum_i \mathbb{E}[\bar{Y}_i]|$.

$$|\frac{1}{n}\sum_i \bar{Y}_i - \frac{1}{n}\sum_i \mathbb{E}[\bar{Y}_i]| \leq O\left(M \cdot \sqrt{\frac{\log(1/\delta)}{n}}\right), \quad w.p. \quad 1 - \delta$$

where it simply follows from Hoeffding's inequality.

**Step 4:** Bound the distance $|\frac{1}{n}\sum_i \bar{Y}_i - \mathbb{E}\left[X_i\right]|$.

$$|\frac{1}{n}\sum_{i\in[n]} \bar{Y}_i - \mathbb{E}\left[X_i\right]| \overset{(a)}{=} |\frac{1}{n}\sum_{i\in\mathcal{G}} \bar{Y}_i - \mathbb{E}\left[X_i\right] + \frac{1}{n}\sum_{i\in\mathcal{B}} \bar{Y}_i|$$

$$\overset{(b)}{\leq} |\frac{1}{n}\sum_{i\in\mathcal{G}} \bar{Y}_i - \mathbb{E}\left[X_i\right]| + 2\alpha M$$

$$= |\frac{1}{n}\sum_{i\in[n]} X_i \mathbb{1}(|X_i| \leq M) - \mathbb{E}\left[X_i\right] - \frac{1}{n}\sum_{i\in[\mathcal{B}]} X_i \mathbb{1}(|X_i| \leq M)| + 2\alpha M$$

$$\leq |\frac{1}{n}\sum_{i\in[n]} X_i \mathbb{1}(|X_i| \leq M) - \mathbb{E}\left[X_i\right]| + 4\alpha M$$

$$\leq \underbrace{|\frac{1}{n}\sum_{i\in[n]} X_i \mathbb{1}(|X_i| \leq M) - \mathbb{E}\left[X_i \mathbb{1}(|X_i| \leq M)\right]|}_{\mathcal{T}_1} + \underbrace{|\mathbb{E}\left[X_i \mathbb{1}(|X_i| \leq M)\right] - \mathbb{E}\left[X_i\right]|}_{\mathcal{T}_2} + 4\alpha M$$

where in (a), $\mathcal{G}$ represents all "good" indexes that are not corrupted and $\mathcal{B}$ represents all "bad" indexes that are corrupted; (b) follows from the boundedness of $\bar{Y}_i$ and the event $\mathcal{E}$ in step 1.

For $\mathcal{T}_2$, by Hölder's inequality and the fact $k$-th moment of $X_i$ is upper bounded by one, we have

$$\mathcal{T}_2 \leq O\left(\frac{1}{M^{k-1}}\right).$$

For $\mathcal{T}_1$, we consider two cases: (i) $k \in (1, 2)$ and (ii) $k \geq 2$ when applying Bernstein's inequality.

For case (i), we note that $\mathbb{E}\left[X_i^2 \mathbb{1}(|X_i| \leq M)\right] = \mathbb{E}\left[|X_i|^k |X_i|^{2-k} \mathbb{1}(|X_i| \leq M)\right] \overset{(a)}{\leq} \mathbb{E}\left[|X_i|^k M^{2-k}\right] \leq M^{2-k}$, where (a) follows from $k < 2$. Thus, by Bernstein's inequality, we have

$$\mathcal{T}_1 \leq O\left(\sqrt{\frac{M^{2-k}\log(1/\delta)}{n}} + \frac{M\log(1/\delta)}{n}\right).$$

For case (ii), we note that $\mathbb{E}\left[X_i^2 \mathbb{1}(|X_i| \leq M)\right] \leq \mathbb{E}\left[X_i^2\right] \leq 1$. Thus, by Bernstein's inequality, we have

$$\mathcal{T}_1 \leq O\left(\sqrt{\frac{\log(1/\delta)}{n}} + \frac{M\log(1/\delta)}{n}\right).$$

**Step 5:** Put everything together. Case (i): for any $k \in (1, 2)$, $\varepsilon \in [0, 1]$, any $\delta \in (0, 1)$ and any $P \in \mathcal{P}_k$, we have with probability at least $1 - \delta$,

$$|\widehat{\mu}_n - \mu(P)| \leq O\left(\sqrt{\frac{M^{2-k}\log(1/\delta)}{n}}\right) + O\left(\frac{M\log(1/\delta)}{n}\right) + O\left(\frac{1}{M^{k-1}}\right) + O(\alpha M) + O\left(\frac{M}{\varepsilon}\cdot\sqrt{\frac{\log(1/\delta)}{n}}\right).$$

Thus, choosing $M = \min\left\{\left(\frac{n}{\log(1/\delta)}\right)^{1/k}, \left(\frac{1}{\alpha}\right)^{1/k}, \left(\frac{\varepsilon\sqrt{n}}{\sqrt{\log(1/\delta)}}\right)^{1/k}\right\}$, yields that

$$|\widehat{\mu}_n - \mu(P)| \leq O\left(\left(\frac{\log(1/\delta)}{n}\right)^{1-1/k} + \alpha^{1-1/k} + \left(\frac{\sqrt{\log(1/\delta)}}{\varepsilon\sqrt{n}}\right)^{1-1/k}\right).$$

Hence, when $n \geq \log(1/\delta)$, we have

$$|\widehat{\mu}_n - \mu(P)| \leq O\left(\left(\frac{\sqrt{\log(1/\delta)}}{\varepsilon\sqrt{n}}\right)^{1-1/k} + \alpha^{1-1/k}\right).$$

Case (ii): for any $k \geq 2$, $\varepsilon \in [0,1]$, any $\delta \in (0,1)$ and any $P \in \mathcal{P}_k$, we have with probability at least $1 - \delta$,

$$|\widehat{\mu}_n - \mu(P)| \leq O\left(\sqrt{\frac{\log(1/\delta)}{n}}\right) + O\left(\frac{M\log(1/\delta)}{n}\right) + O\left(\frac{1}{M^{k-1}}\right) + O(\alpha M) + O\left(\frac{M}{\varepsilon} \cdot \sqrt{\frac{\log(1/\delta)}{n}}\right).$$

Thus, choosing $M = \min\left\{\left(\frac{n}{\log(1/\delta)}\right)^{1/k}, \left(\frac{1}{\alpha}\right)^{1/k}, \left(\frac{\varepsilon\sqrt{n}}{\sqrt{\log(1/\delta)}}\right)^{1/k}\right\}$, yields that

$$|\widehat{\mu}_n - \mu(P)| \leq O\left(\sqrt{\frac{\log(1/\delta)}{n}} + \left(\frac{\log(1/\delta)}{n}\right)^{1-1/k} + \alpha^{1-1/k} + \left(\frac{\sqrt{\log(1/\delta)}}{\varepsilon\sqrt{n}}\right)^{1-1/k}\right).$$

Hence, when $n \geq \log(1/\delta)$, we have

$$|\widehat{\mu}_n - \mu(P)| \leq O\left(\left(\frac{\sqrt{\log(1/\delta)}}{\varepsilon\sqrt{n}}\right)^{1-1/k} + \alpha^{1-1/k}\right).$$

Finally, combining the above two cases, we see that when $n \geq \log(1/\delta)$, for any $k > 1$, it suffices to choose $M = \min\left\{\left(\frac{1}{\alpha}\right)^{1/k}, \left(\frac{\varepsilon\sqrt{n}}{\sqrt{\log(1/\delta)}}\right)^{1/k}\right\}$ and obtain that

$$|\widehat{\mu}_n - \mu(P)| \leq O\left(\left(\frac{\sqrt{\log(1/\delta)}}{\varepsilon\sqrt{n}}\right)^{1-1/k} + \alpha^{1-1/k}\right).$$

which finishes the proof for the CTL setting.

# G  Proof of the Upper Bound for the C-LDP-C Setting

After the proofs for the previous two settings, we can easily establish the upper bound for the C-LDP-C setting. For completeness, we also provide a detailed proof. To elucidate the utility of our approach, we will structure the proof into four distinct steps, building upon the derivation outlined previously.

$$X_i \xrightarrow{\text{Corruption}} Y_i \xrightarrow{\text{Trunc.}(M)} \bar{Y}_i \xrightarrow{\text{Random Rounding}} Y_i' \xrightarrow{\text{Random Response}} Z_i \xrightarrow{\text{Corruption}} Z_i' \xrightarrow{\text{Trunc.}(M\frac{e^\varepsilon+1}{e^\varepsilon-1})} \bar{Z}_i \xrightarrow{\text{Sample Mean}} \widehat{\mu}_n$$

**Step 1:** Bound the distance $|\mathbb{E}[Y_i'] - \widehat{\mu}_n|$.
According to the analysis in the LTC setting, we can directly derive

$$|\widehat{\mu}_n - \mathbb{E}[Y_i']| = \left|\frac{1}{n}\sum_i \bar{Z}_i - \mathbb{E}[Y_i']\right|$$

$$\leq 2\alpha \cdot M \cdot \frac{e^\varepsilon+1}{e^\varepsilon-1} + O\left(M \cdot \frac{e^\varepsilon+1}{e^\varepsilon-1} \cdot \sqrt{\frac{\log(1/\delta)}{n}}\right) \quad w.p. \quad 1-\delta$$

**Step 2:** Bound the distance $|\frac{1}{n}\sum_i \bar{Y}_i - \frac{1}{n}\sum_i \mathbb{E}[\bar{Y}_i]|$.

$$|\frac{1}{n}\sum_i \bar{Y}_i - \frac{1}{n}\sum_i \mathbb{E}[\bar{Y}_i]| \leq O\left(M \cdot \sqrt{\frac{\log(1/\delta)}{n}}\right), \quad w.p. \quad 1-\delta$$

where it simply follows from Hoeffding's inequality.

**Step 3:** Bound the distance $|\frac{1}{n}\sum_i \bar{Y}_i - \mathbb{E}[X_i]|$.
From the analysis in the CTL setting, we once again derive

$$|\frac{1}{n}\sum_{i\in[n]} \bar{Y}_i - \mathbb{E}[X_i]| \leq \underbrace{|\frac{1}{n}\sum_{i\in[n]} X_i \mathbb{1}(|X_i| \leq M) - \mathbb{E}[X_i \mathbb{1}(|X_i| \leq M)]|}_{\mathcal{T}_1} + \underbrace{|\mathbb{E}[X_i \mathbb{1}(|X_i| \leq M)] - \mathbb{E}[X_i]|}_{\mathcal{T}_2} + 4\alpha M$$

**Step 4:** Put everything together.

Case (i): for any $k \in (1, 2)$, $\varepsilon \in [0, 1]$, any $\delta \in (0, 1)$ and any $P \in \mathcal{P}_k$, we have with probability at least $1 - \delta$,

$$|\widehat{\mu}_n - \mu(P)| \leq O\left(\sqrt{\frac{M^{2-k}\log(1/\delta)}{n}}\right) + O\left(\frac{M\log(1/\delta)}{n}\right) + O\left(\frac{1}{M^{k-1}}\right)$$
$$+ O(\alpha M) + O(\frac{\alpha M}{\varepsilon}) + O\left(\frac{M}{\varepsilon} \cdot \sqrt{\frac{\log(1/\delta)}{n}}\right).$$

Thus, choosing $M = \min\left\{\left(\frac{n}{\log(1/\delta)}\right)^{1/k}, \left(\frac{\varepsilon}{\alpha}\right)^{1/k}, \left(\frac{\varepsilon\sqrt{n}}{\sqrt{\log(1/\delta)}}\right)^{1/k}\right\}$, yields that

$$|\widehat{\mu}_n - \mu(P)| \leq O\left(\left(\frac{\log(1/\delta)}{n}\right)^{1-1/k} + \left(\frac{\alpha}{\varepsilon}\right)^{1-1/k} + \left(\frac{\sqrt{\log(1/\delta)}}{\varepsilon\sqrt{n}}\right)^{1-1/k}\right).$$

Hence, when $n \geq \log(1/\delta)$, we have

$$|\widehat{\mu}_n - \mu(P)| \leq O\left(\left(\frac{\sqrt{\log(1/\delta)}}{\varepsilon\sqrt{n}}\right)^{1-1/k} + \left(\frac{\alpha}{\varepsilon}\right)^{1-1/k}\right).$$

Case (ii): for any $k \geq 2$, $\varepsilon \in [0, 1]$, any $\delta \in (0, 1)$ and any $P \in \mathcal{P}_k$, we have with probability at least $1 - \delta$,

$$|\widehat{\mu}_n - \mu(P)| \leq O\left(\sqrt{\frac{\log(1/\delta)}{n}}\right) + O\left(\frac{M\log(1/\delta)}{n}\right) + O\left(\frac{1}{M^{k-1}}\right)$$
$$+ O(\alpha M) + O(\frac{\alpha M}{\varepsilon}) + O\left(\frac{M}{\varepsilon} \cdot \sqrt{\frac{\log(1/\delta)}{n}}\right).$$

Thus, choosing $M = \min\left\{\left(\frac{n}{\log(1/\delta)}\right)^{1/k}, \left(\frac{\varepsilon}{\alpha}\right)^{1/k}, \left(\frac{\varepsilon\sqrt{n}}{\sqrt{\log(1/\delta)}}\right)^{1/k}\right\}$, yields that

$$|\widehat{\mu}_n - \mu(P)| \leq O\left(\sqrt{\frac{\log(1/\delta)}{n}} + \left(\frac{\log(1/\delta)}{n}\right)^{1-1/k} + \left(\frac{\alpha}{\varepsilon}\right)^{1-1/k} + \left(\frac{\sqrt{\log(1/\delta)}}{\varepsilon\sqrt{n}}\right)^{1-1/k}\right).$$

Hence, when $n \geq \log(1/\delta)$, we have

$$|\widehat{\mu}_n - \mu(P)| \leq O\left(\left(\frac{\sqrt{\log(1/\delta)}}{\varepsilon\sqrt{n}}\right)^{1-1/k} + \left(\frac{\alpha}{\varepsilon}\right)^{1-1/k}\right).$$

Finally, combining the above two cases, we see that when $n \geq \log(1/\delta)$, for any $k > 1$, it suffices to choose $M = \min\left\{\left(\frac{\varepsilon}{\alpha}\right)^{1/k}, \left(\frac{\varepsilon\sqrt{n}}{\sqrt{\log(1/\delta)}}\right)^{1/k}\right\}$ and obtain that

$$|\widehat{\mu}_n - \mu(P)| \leq O\left(\left(\frac{\sqrt{\log(1/\delta)}}{\varepsilon\sqrt{n}}\right)^{1-1/k} + \left(\frac{\alpha}{\varepsilon}\right)^{1-1/k}\right).$$

$\square$

# H    Proof of Proposition 3

*Proof.* As in the section for mean estimation, we first focus on the LTC setting and divide the lower bound proof into two steps.

**Step 1: Without corruption.** In this case, we aim to establish the second term in the lower bound. We consider the first MAB instance $I$ as follows. Let $\gamma > 0$ be determined later and

$$P_1(X = 1/\gamma) = 1/2 \cdot \gamma^k, \quad P_1(X = 0) = 1 - 1/2 \cdot \gamma^k$$

$$P_a(X = 1/\gamma) = 1/4 \cdot \gamma^k, \quad P_a(X = 0) = 1 - 1/4 \cdot \gamma^k. \quad \forall a \neq 1. \tag{7}$$

Thus, one can see that $I \in \mathrm{MAB}(k)$ for a proper choice of $\gamma$ and arm 1 is the optimal arm for instance $I$. We let $M_a$ be the induced marginal distribution of $P_a$ via any $\varepsilon$-LDP channel and $\mathbb{E}_I[\cdot]$ denote the expectation over $\mathbb{P}_I$, which is over the randomness in the marginal distributions $\{M_a\}_{a \in [K]}$ and policy $\pi$.

Then, we construct a "coupled" instance $I'$ of $I$ as follows. Let $i = \arg\min_{j>1} \mathbb{E}_I[N_j(T)]$, i.e., the arm between $a_2$ and $a_K$ that has the minimum number of pulls under instance $I$. Define the second instance $I'$ that only differs in the distribution for arm $i$ compared to instance $I$

$$P_i(X = 1/\gamma) = 3/4 \cdot \gamma^k, \quad P_i(X = 0) = 1 - 3/4 \cdot \gamma^k. \tag{8}$$

Thus, $I' \in \mathrm{MAB}(k)$ and arm $i$ is the optimal arm for instance $I'$. By definition, we also have $\mathbb{E}_I[N_i(T)] \leq T/(K-1)$.

For any instance $I$ and policy $\pi$, we let $\mathcal{R}_T(\pi, I)$ be its corresponding expected regret. Then, by standard argument and noting that the mean gap is $\Delta := 1/4 \cdot \gamma^{k-1}$, we have

$$
\begin{aligned}
\mathcal{R}_T(\pi, I) + \mathcal{R}_T(\pi, I') &\geq \frac{T}{2} \cdot \Delta \cdot (\mathbb{P}_I[N_1(T) \leq T/2] + \mathbb{P}_{I'}[N_1(T) \geq T/2]) \\
&\overset{(a)}{\geq} \frac{T\Delta}{4} \exp(-\mathrm{KL}(\mathbb{P}_I \| \mathbb{P}_{I'})) \\
&\overset{(b)}{=} \frac{T\Delta}{4} \exp(-\mathbb{E}_I[N_i(T)] \cdot \mathrm{KL}(M_i \| M_i')) \\
&\overset{(c)}{\geq} \frac{T\Delta}{4} \exp(-\mathbb{E}_I[N_i(T)] \cdot 4(e^\varepsilon - 1)^2 \cdot (\mathrm{TV}(P_i, P_i'))^2) \\
&\overset{(d)}{\geq} \frac{T\Delta}{4} \exp\left(-\frac{T}{K-1} \cdot 4(e^\varepsilon - 1)^2 \cdot (\mathrm{TV}(P_i, P_i'))^2\right) \\
&\overset{(e)}{=} \frac{T\Delta}{4} \exp\left(-\frac{T}{K-1} \cdot 4(e^\varepsilon - 1)^2 \cdot \frac{\gamma^{2k}}{4}\right)
\end{aligned}
$$

where (a) holds by Bretagnolle–Huber inequality; (b) follows from chain rule of KL divergence; (c) holds by Theorem 1 in Duchi et al. (2018); (d) is true since $\mathbb{E}_I[N_i(T)] \leq T/(K-1)$; (e) holds by definition of TV distance.

Thus, putting everything together and noting that for $\varepsilon \in [0,1]$, $e^\varepsilon - 1 \leq 2\varepsilon$, yields that

$$\mathcal{R}_T(\pi, I) + \mathcal{R}_T(\pi, I') \geq \frac{T\Delta}{4} \exp\left(-4\frac{\varepsilon^2 T \gamma^{2k}}{K-1}\right).$$

Thus, suppose $T \geq K/\varepsilon^2$ and choosing $\gamma = (K/(\varepsilon^2 T))^{1/2k}$, one can check that all the required conditions on $\gamma$ are satisfied and we finally have that $\max\{\mathcal{R}_T(\pi, I), \mathcal{R}_T(\pi, I')\} \geq \Omega(T\gamma^{k-1}) = \Omega\left(T^{\frac{k+1}{2k}}\left(\frac{K}{\varepsilon^2}\right)^{\frac{k-1}{2k}}\right)$.

**Step 2: Corruption part.** In this case, we aim to establish the first term in the lower bound.

This part basically shares the same argument as before for mean estimation. Note that the only difference between $I$ and $I'$ is the distribution for arm $i$. Then, we apply the same argument as in the proof of Proposition 1 to $P_i$ and $P_i'$. Hence, we have that there exists Huber corruptions so that one cannot distinguish between $P_i$ and $P_i'$, and hence $I$ and $I'$. As a result, the total expected regret is $\Omega(T\gamma^{k-1}) = \Omega(T(\alpha/\varepsilon)^{1-1/k})$.

Finally, for the CTL setting, the first step is the same and second step only differs in that there is no "contraction" effect as in the proof of Proposition 1. □

# I Proof of Proposition 4

*Proof.* Let us start with the LTC case. We divide the set of all sub-optimal arms $\mathcal{G}$ into two groups $\mathcal{G}_1$ and $\mathcal{G}_2 := \mathcal{G} \setminus \mathcal{G}_1$, where $\mathcal{G}_1 = \{a \in [K] \setminus a^* : c' \left(\frac{\alpha}{\varepsilon}\right)^{1-1/k} < \frac{1}{2}\Delta_a\}$ for some universal constant $c'$ chosen later. Hence, $\mathcal{G}_2 = \{a \in [K] \setminus a^* : c' \left(\frac{\alpha}{\varepsilon}\right)^{1-1/k} \geq \frac{1}{2}\Delta_a\}$, which implies that the total expected regret from suboptimal arms in $\mathcal{G}_2$ is upper bounded by $O \left(T \left(\frac{\alpha}{\varepsilon}\right)^{1-1/k}\right)$. Thus, it remains to bound the total expected regret of pulling suboptimal arms in $\mathcal{G}_1$. To this end, for each $i \in \mathcal{G}_1$, we aim to show that

$$\mathbb{E}\left[N_i(T)\right] \leq O \left(\frac{\log T}{\varepsilon^2 (\Delta_i)^{\frac{2k}{k-1}}} + \frac{\log T}{\alpha}\right). \tag{9}$$

Let us first assume (9) holds and see how we can arrive at our claimed upper bound. By the definition of expected regret, we have

$$
\begin{aligned}
R(k, \varepsilon, \alpha, T) &= \sum_{i \in \mathcal{G}_1} \Delta_i \mathbb{E}\left[N_i(T)\right] + \sum_{i \in \mathcal{G}_2} \Delta_i \mathbb{E}\left[N_i(T)\right] \\
&\leq \sum_{i \in \mathcal{G}_1} \Delta_i \mathbb{E}\left[N_i(T)\right] + O\left(T\left(\frac{\alpha}{\varepsilon}\right)^{1-1/k}\right),
\end{aligned}
$$

where inequality holds by the definition of $\mathcal{G}_2$. It remains to translate the first term into a problem-independent one. To this end, we further divide the arms in $\mathcal{G}_1$ into two groups: one group consists of all arms that satisfy $\Delta_i < \eta$ for some constant $\eta > 0$ and another one contains all arms that satisfy $\Delta_i \geq \eta$. Thus, by (9), we have

$$\sum_{i \in \mathcal{G}_1} \Delta_i \mathbb{E}\left[N_i(T)\right] \leq \eta T + O \left(\frac{K \log T}{\varepsilon^2 \eta^{\frac{k+1}{k-1}}} + \frac{K \log T}{\alpha}\right).$$

Choosing $\eta = \left(\frac{K \log T}{\varepsilon^2 T}\right)^{\frac{k-1}{2k}}$, yields that the total expected regret satisfies

$$R(k, \varepsilon, \alpha, T) \leq O \left(\left(\frac{K \log T}{\varepsilon^2}\right)^{\frac{k-1}{2k}} T^{\frac{k+1}{2k}} + \frac{K \log T}{\alpha} + T \left(\frac{\alpha}{\varepsilon}\right)^{1-1/k}\right).$$

Finally, for very small $\alpha$, one can replace it with its upper bound $\bar{\alpha}$ to optimize the regret.

It remains to establish (9). First note that $O(\log T / \alpha)$ basically follows from the burn-in period. Thus, we only need to bound the total number of pulls after the burn-in period. We denote by $N_i'(t)$ the total number of by time $t$ after the burn-in period, i.e., it is equal to $N_i(t)$ minus the total number of burn-in plays of arm $i$. In the following, we will show that

$$\mathbb{E}\left[N_i'(T)\right] \leq C_1 \frac{\log T}{\varepsilon^2 (\Delta_i)^{\frac{2k}{k-1}}} + C_2, \tag{10}$$

for some constants $C_1$ and $C_2$.

To this end, for $t$ that is after the burn-in period of arm $i \in \mathcal{G}_1$, if $a_t = i$, then one of the following must be true:

$$\text{UCB}_{a^*}(t) \leq \mu(P_{a^*}) \tag{11}$$

$$\widehat{\mu}_{i,N_i(t)} > \mu(P_i) + c \left(\frac{\alpha}{\varepsilon}\right)^{1-1/k} + c \left(\frac{1}{\varepsilon}\sqrt{\frac{\log(t^4)}{N_i(t)}}\right)^{1-1/k} \tag{12}$$

$$N_i'(t) < C \frac{\log T}{\varepsilon^2 (\Delta_i)^{\frac{2k}{k-1}}} \tag{13}$$

This is because if all three are not true, then we have

$$
\begin{aligned}
\mathrm{UCB}_{a^\star}(t) &> \mu(P_{a^\star}) \\
&= \mu(P_i) + \Delta_i \\
&\overset{(a)}{\geq} \mu(P_i) + \frac{1}{2}\Delta_i + c'\left(\frac{\alpha}{\varepsilon}\right)^{1-1/k} \\
&\overset{(b)}{\geq} \mu(P_i) + 2c\left(\frac{1}{\varepsilon}\sqrt{\frac{\log(t^4)}{N_i'(t)}}\right)^{1-1/k} + c'\left(\frac{\alpha}{\varepsilon}\right)^{1-1/k} \\
&\overset{(c)}{\geq} \mu(P_i) + 2c\left(\frac{1}{\varepsilon}\sqrt{\frac{\log(t^4)}{N_i(t)}}\right)^{1-1/k} + c'\left(\frac{\alpha}{\varepsilon}\right)^{1-1/k} \\
&\overset{(d)}{\geq} \widehat{\mu}_{i,N_i(t)} + c\left(\frac{1}{\varepsilon}\sqrt{\frac{\log(t^4)}{N_i(t)}}\right)^{1-1/k} + c\left(\frac{\alpha}{\varepsilon}\right)^{1-1/k} \\
&= \mathrm{UCB}_i(t)
\end{aligned}
$$

where (a) holds by the fact that $i \in \mathcal{G}_1$; (b) holds by choosing a large constant $C$ in (13); (c) is true since $N_i(t) > N_i'(t)$; (d) holds by the inverse direction of (12) and choosing $c' = 2c$.

Let $t'$ be the time just after the burn-in period, then we have

$$
\begin{aligned}
\mathbb{E}\left[N_i'(T)\right] = \mathbb{E}\left[\sum_{t\geq t'} \mathbb{1}(a_t = i)\right] &\leq C\frac{\log T}{\varepsilon^2(\Delta_i)^{\frac{2k}{k-1}}} + \sum_{t\geq t'}\mathbb{E}\left[\mathbb{1}(a_t = i \text{ and } (13) \text{ is false})\right] \\
&\overset{(a)}{\leq} C\frac{\log T}{\varepsilon^2(\Delta_i)^{\frac{2k}{k-1}}} + \sum_{t\geq t'}\mathbb{E}\left[\mathbb{1}((11) \text{ is true or } (12) \text{ is true})\right]
\end{aligned}
$$

where (a) holds by the above claim, i.e., if $a_t = i$ and (13) is false, then one of (11) and (12) must be true. Then, by our mean concentration result and union bounds, we can upper bound the second term above as

$$
\mathbb{E}\left[\mathbb{1}((11) \text{ is true or } (12) \text{ is true})\right] \leq 2\sum_{s=1}^{t}\frac{1}{t^4} = \frac{2}{t^3}.
$$

Putting them together, we have established (10), hence the result. The proof for CTL setting is essentially the same with the only difference in the definition of $\mathcal{G}_1$ and $\mathcal{G}_2$. $\square$

## J  Proof of Proposition 5

*Proof.* **Without corruption.** We consider two instances in $\mathrm{MAB}(\beta^\star, k)$. In particular, we consider two-arm MABs $I$ and $I'$:

$$
\begin{aligned}
\text{For } I : \mu_1^I &:= \mu(P_1^I) = 1/2 \cdot \gamma^{k-1}, \ \mu_2^I := \mu(P_2^I) = 1/4 \cdot \gamma^{k-1} \\
\text{For } I' : \mu_1^{I'} &:= \mu(P_1^I) = 1/2 \cdot \gamma^{k-1}, \ \mu_2^{I'} := \mu(P_2^I) = 3/4 \cdot \gamma^{k-1}
\end{aligned}
\tag{14}
$$

These distributions can be constructed in the same way as in the proof of Proposition 3 (cf. (7)). Moreover, for the behavior policy $\pi$, we have $\pi(2) = 1/\beta^\star$ and $\pi(1) = 1 - 1/\beta^\star$. We now verify that both $(\pi, \mu^I)$ and $(\pi, \mu^{I'})$ are in $\mathrm{MAB}(\beta^\star, k)$. By construction, each distribution is belonging to $\mathcal{P}_k$. It remains to verify that $1/\pi(a^\star) \leq \beta^\star$. For $I'$, we have $1/\pi(2) = \beta^\star$. And for $I$, we have $1/\pi(1) = 1/(1 - 1/\beta^\star) \leq \beta^\star$ when $\beta^\star \geq 2$.

Now, we proceed to apply classic Le Cam's method. Let loss/sub-optimality of any final chosen arm $\widehat{a} \in \{1, 2\}$ under $I$ and $I'$ be $\ell(\widehat{a}; I), \ell(\widehat{a}; I')$. Then, by our construction, we have

$$
\ell(\widehat{a}; I) + \ell(\widehat{a}; I') \geq 1/4 \cdot \gamma^{k-1}.
$$

Thus, by Le Cam's method and Bretagnolle–Huber inequality, we have

$$\text{SubOpt}^*(\beta^\star, k, \varepsilon, \alpha, N) \geq \frac{\gamma^{k-1}}{16} \exp\left(-\text{KL}\left(M_\pi^I \| M_\pi^{I'}\right)\right),$$

where $\text{KL}\left(M_\pi^I \| M_\pi^{I'}\right)$ is the private KL divergence between two MAB instances. By chain rule of KL divergence and Theorem 1 in Duchi et al. (2018), we have

$$\text{KL}\left(M_\pi^I \| M_\pi^{I'}\right) \leq \frac{N}{\beta^\star} 4(e^\varepsilon - 1)^2 \left(\text{TV}\left(P_2^I, P_2^{I'}\right)\right)^2.$$

Thus, noting that for $\varepsilon \in [0,1]$, $e^\varepsilon - 1 \leq 2\varepsilon$ and $\left(\text{TV}\left(P_2^I, P_2^{I'}\right)\right)^2 = \frac{\gamma^{2k}}{4}$, we have that

$$\text{SubOpt}^*(\beta^\star, k, \varepsilon, \alpha, N) \geq \frac{\gamma^{k-1}}{16} \exp\left(-\frac{\varepsilon^2 N \gamma^{2k}}{4\beta^\star}\right).$$

Finally, for a large enough $N$, choosing $\gamma = (\beta^\star/(\varepsilon^2 N))^{1/2k}$, yields that

$$\text{SubOpt}^*(\beta^\star, k, \varepsilon, \alpha, N) \geq \Omega\left(\left(\frac{1}{\varepsilon}\sqrt{\frac{\beta^\star}{N}}\right)^{1-1/k}\right).$$

**Corruption part.** By our construction (cf. (14) (7), (8)) we have that $\text{TV}\left(P_2^I, P_2^{I'}\right) = \frac{\gamma^k}{2}$. Then, a similar idea as in the proof of Proposition 1 applies here. That is, for the LTC setting, by the contraction of LDP, we can set $\gamma^k = \Theta(\frac{\alpha}{\varepsilon})$ so that $\text{TV}\left(M_2^I, M_2^{I'}\right) \leq \alpha$. Thus, one cannot distinguish $I$ and $I'$ under $\alpha$-Huber model. Thus, one has to incur a sub-optimality gap as $\Omega(\gamma^k) = \left(\frac{\alpha}{\varepsilon}\right)^{1-1/k}$. In contrast, due to no contraction by LDP first, one can only set $\gamma^k = \Theta(\alpha)$, which leads to the final result. $\qquad\square$

# K  Proof of Proposition 6

*Proof.* We will focus on the LTC case, since the CTL case is nearly the same with a minor change in the confidence bound. Let $\mathcal{E} = \mathcal{E}_1 \cap \mathcal{E}_2$ where

$$\mathcal{E}_1 := \{\forall a \in [K], |\widehat{r}(a) - r(a)| \leq b(a)\}$$

$$\mathcal{E}_2 := \{N(a^\star) \geq \frac{1}{2}N\pi(a^\star)\}.$$

Let us first assume that $\mathbb{P}[\mathcal{E}] \geq 1 - 2\delta$ and see how we can prove the final result. Then, we will establish this high-probability event in the end. Hence, condition the event $\mathcal{E}$ and define $\text{LCB}(a) := \widehat{r}(a) - b(a)$, we have

$$r(a^\star) - r(\widehat{a}) = r(a^\star) - \text{LCB}(a^\star) + \text{LCB}(a^\star) - \text{LCB}(\widehat{a}) + \text{LCB}(\widehat{a}) - r(\widehat{a})$$
$$\leq 2b(a^\star)$$

$$\leq 2c\left(\frac{\alpha}{\varepsilon}\right)^{1-1/k} + 2c\left(\frac{1}{\varepsilon}\sqrt{\frac{\log(2K/\delta)}{N_{a^\star}}}\right)^{1-1/k}.$$

Then, by the definition of $\beta^\star$ and $\mathcal{E}$, we can further lower bound $N_{a^\star}$ by $\frac{N}{2\beta^\star}$. Then, by the bounded mean of each arm, and choosing $\delta = 1/N$, we have the claimed expected sub-optimality result.

It remains to bound the probability of $\mathcal{E}$. For $\mathcal{E}_2$, by standard Chernoff bound, we have $\mathbb{P}[\mathcal{E}_2] \geq 1 - \delta$ when $N \geq 8\beta^\star \log(1/\delta)$. For $\mathcal{E}_1$, we have the following argument. For any arm $a$ such that $N_a$ is larger than the burn-in threshold, the concentration in $\mathcal{E}_1$ follows from our high-probability mean estimation result. For all other arms, by construction and the condition that all arms have mean between $[-1, 1]$, we have

$$\widehat{r}(a) - b(a) = -1 \leq r(a) \leq \widehat{r}(a) + b(a) = 1,$$

which enables us to establish our claim $\mathbb{P}[\mathcal{E}] \geq 1 - 2\delta$.

$\qquad\square$

