# OpenReview forum: "Locally Private and Robust Multi-Armed Bandits"
_NeurIPS.cc/2024/Conference — NeurIPS 2024 poster_

### Official Review · Reviewer_fKk6 · 2024-07-11

**Soundness:** 3
**Presentation:** 3
**Contribution:** 3
**Rating:** 7
**Confidence:** 3

**Summary:**

The study examines the interaction between local differential privacy (LDP) and robustness to Huber corruption and heavy-tailed rewards in multi-armed bandits (MABs). It focuses on two scenarios: LDP-then-Corruption (LTC) and Corruption-then-LDP (CTL). They provide a tight characterization of mean estimation error and minimax regret in both scenarios, supported by simulations. The findings indicate LTC results in worse performance than CTL. Additionally, the study offers the first performance bounds for scenarios with corruption both before and after LDP, and corrects previous errors in regret bounds for locally private, heavy-tailed online MABs

**Strengths:**

This model considering both local differential privacy (LDP) and robustness to Huber corruption is new and they provide a unified algorithm for online MAB with better minimax upper bound than previous works. They also propose the minimax lower bound for the first time.

**Weaknesses:**

The MAB algorithms still involve the hyper-parameter tuning.

**Questions:**

Q1: You mention that the upper bound in Tao's paper is incorrect, can you further point out which step in their proof is wrong?

Q2: What is the novelty of your mean estimation in Algorithm 1?

**Limitations:**

For online MAB, the paper just considers the UCB structure, maybe also consider the extension for Thompson Sampling will make the story more complete.

---

> ### Author Rebuttal · Authors · 2024-08-05
>
> **Technical flaw in Tao et al. 2022.** Sure, we have provided a detailed discussion on it, which points out the incorrect step in their upper bound proof (see the above general response).
>
> **Novelty of Algorithm 1.** We remark on the novelty (or insight) of Algorithm 1 from the perspectives of algorithmic design and analysis, respectively.
>
> 1. Algorithmic design: We use random response for privacy rather than the Laplace mechanism. This is due to the key observation that under corruption and LDP, the Laplace mechanism no longer gives the optimal rate, highlighting the interesting interplay of corruption and LDP.
>
> 2. Analysis: One key novelty in our analysis is that Algorithm 1 can adaptively give optimal rates for all three different scenarios (LTC, CTL, and C-LDP-C) for bounded data.
>
> **Hyper-parameter tuning.** Yes, our MAB algorithms need to determine the optimistic or pessimistic radius, which requires us to know $\epsilon$, $\alpha$, $k$, and some constant $c$. We first note that even in non-private, non-corrupted cases, the radius also requires tuning with respect to $c$ in practical implementations. Regarding other parameters (in particular $\alpha$), we tend to believe that it is necessary to be known in advance to guarantee optimal rates. Finally, to achieve the optimal rate in CTL, one has to carefully choose the right radius, while C-LDP-C (corruption-LDP-corruption) can be the same as LTC.
>
>
> **Other exploration strategies.** In this paper, we use UCB as an example. Since the key step in other variants (like Thompson sampling) also relies on a tight mean estimation, we believe one can generalize our results to other exploration strategies.

---

> > ### Comment · Reviewer_fKk6 · 2024-08-13
> >
> > Thanks for your rebuttal and I will keep my score.

---

> > > ### Author Response · Authors · 2024-08-13
> > > **Thank you**
> > >
> > > Thanks again for your comments and positive evaluation of our paper!

---

### Official Review · Reviewer_5Gx9 · 2024-07-12

**Soundness:** 3
**Presentation:** 3
**Contribution:** 2
**Rating:** 6
**Confidence:** 3

**Summary:**

This paper studies multi-armed bandits where the feedback is a locally differentially private and corrupted version of the true rewards. The main message is that the order in which the rewards are (1) corrupted and (2) made private (i.e., (1) and then (2) or (2) and then (1)) changes the achievable regret rates, which they sharply characterize with new matching lower and upper bounds.

**Strengths:**

* New and sharp regret upper and lower regret bounds for this locally private and corrupt bandit setting.
* Tight characterization of private and corrupt mean estimation error.
* There is expansive coverage of prior works on DP bandits, which helps place this contribution in context of the literature.

**Weaknesses:**

* I do think the writing is repetitive at times, and the paper could be better served by simplifying exposition and moving focus to key intuitions (for instance, Propositions 1 and 2 seem redundant given Theorem 1). Additionally, there are many settings and problems (LTC, CTL, C-LDP-C), online and offline MABs, so a summary table or glossary could be helpful to get the high level sense of the results.
* It is a bit difficult to get the sense of novelties over prior works (Tao et al., 2022; Wu et al., 2023)). For instance, the local DP model is stronger than central DP, but what is the new technical difficulty handled in this setting? Also, there is not really much discussion of what the mistake of the previous state-of-art Tao et al., 2022 is other than that it contradicts the results of this submission. It would be easier for a quick reader to get a sense of confidence if the exact mistake in analysis could be elaborated. Because of the elaborate setting of this paper (i.e., local privacy, corruption, heavy-tail reward, ordering of corruption vs. privacy) and it seems various slices or weaker versions of this specifically arranged setting have been studied in other works, it's difficult to get a sense if the paper is not just a combinatorial composition of prior approaches for this setting or if some important technical obstacle was truly overcome to obtain the results.
* Although I understand this is mostly a theory paper, I think it would also be good to have some practical discussion of when LTC or CTL might appear in application and what kind of implications the "LTC is harder" message has. Likewise, what are application scenarios for LTC or CTL?

**Questions:**

Please see weaknesses above.

**Limitations:**

No broader impact concern.

---

> ### Author Rebuttal · Authors · 2024-08-05
>
> We thank the reviewer for your time and review. Below is our response to the comments.
>
> **Presentation.** Thanks for the suggestion on the presentation. We will try to incoprate them in the final version, e.g., a summary table.
>
> **Practical scenarios of LTC and CTL** Yes, we have already briefly discussed them in Appendix C.1.
>
> **Important contributions compared to previous works.** Our result cannot be obtained by directly following previous works.
>
> 1. Even without corruption, the state-of-the-art result is ungrounded. We discuss it further above (see the general response), which highlights the incorrect step in the upper bound proof of Tao et al. 2022. This complements the points we have made from the lower bound perspective.
>
> 2. In contrast to Wu et al. 2023's setting of corruption plus central DP, where the standard Laplace mechanism will work, we have shown that the Laplace mechanism will only give a sub-optimal rate in the setting of local DP plus corruption. See Remark 4.

---

> > ### Comment · Reviewer_5Gx9 · 2024-08-14
> >
> > Thank you for your response and clarifying the mistake in the previous work. My concerns are addressed and I raise my score.

---

### Official Review · Reviewer_1jNj · 2024-07-17

**Soundness:** 3
**Presentation:** 3
**Contribution:** 3
**Rating:** 6
**Confidence:** 4

**Summary:**

This paper considers the heavy-tailed online and offline MAB problem with local differential privacy and Huber corruption, where the CTL, LTC, and C-LDP-C models are considered. By first providing tight high-probability concentration bounds for the mean-estimation problem under the CTL and LTC settings, the authors offer new upper bound results for online and offline MAB. Lower bound results are also established to demonstrate the tightness of the upper bound results.

**Strengths:**

The presentation of results in this paper is clear, and the comments are detailed. Related works are well discussed. Several novel results are established and well generalize the previous LDP heavy-tailed estimation results, including:

1. New high-probability bounds for LDP heavy-tailed and corrupted mean estimation problems.

2. An interesting separation result between the CTL and LTC settings, with detailed explanations.

3. New results for offline and online MAB.

**Weaknesses:**

Several claims are not so clear to me and may need further explanation and clarification, including:

1. In the regret bound for online MAB, it seems that the upper bound result in Proposition 4 cannot directly imply the result claimed in Theorem 2. I take the CTL result as an example, and the LTC result has the same problem:
In Theorem 2, the upper bound reads as $O(T({\alpha}/\varepsilon)^{1-1/k})$.
In Proposition 4, there is a factor $\tilde{\alpha} \in [\alpha, 1/2]$ that balances the trade-off $\tilde{O}( T(\tilde{\alpha}/\varepsilon)^{1-1/k} + 1/\tilde{\alpha})$. In particular, when $1/\alpha > T({\alpha}/\varepsilon)^{1-1/k}$, it seems that Proposition 4 can no longer recover Theorem 2?

2. In the offline MAB result, while authors claimed that the upper bound result almost matches the proposed lower bound, it seems that the dependency on $N$ is not optimal when comparing the lower bound term $\sqrt{1/N}$ with the upper bound result $(1/N)^{1/2 - 1/2k}$?

**Questions:**

1.For the questions regarding the claim of the theorems, see Weakness 1 and 2.

2.As discussed in C.3, authors states that the previous SOTA in the heavy-tailed LDP MAB paper [1] has a technical flaw. The authors have demonstrated this claim by showing that the upper bound result in [1] can even break the lower bound result proved in this paper. I am wondering, besides this evidence, can the authors explicitly point out any technical flaw in the proof of the result in [1]?

[1] Youming Tao, Yulian Wu, Peng Zhao, and Di Wang. Optimal rates of (locally) differentially private heavy-tailed multi-armed bandits. In International Conference on Artificial Intelligence and Statistics, pp. 1546–1574. PMLR, 2022.

**Limitations:**

N.A.

---

> ### Author Rebuttal · Authors · 2024-08-05
>
> We thank the reviewer for your time and review. We will provide our response below.
>
> **Online MAB.** In this paper, as in previous work (e.g., Wu et al 2023), we treat $\alpha$ as a constant (i.e., it does not scale with $T$). Thus, the third term in big-O of Proposition 4 is a lower order term as $T \to \infty$, which gives us Theorem 2.
>
> **Offline MAB.** There is a **typo** in Proposition 5, i.e., the exponent term of $1-1/k$ is missing. Our proof in Appendix J indeed has the right one, which matches the upper bound in Proposition 6. The updated lower bounds in Proposition 5 are shown below:
>
> (i) LTC: $\mathrm{SubOpt}^{\ast}_{\mathrm{LTC}}(\beta^{\ast}, k, \epsilon, \alpha, N) \geq \Omega\left(\left(\frac{\alpha}{\epsilon}\right)^{1-1/k} + \left(\frac{1}{\epsilon} \sqrt{\frac{\beta^{\ast}}{N}}\right)^{1-1/k}\right)$
>
> (ii) CTL: $\mathrm{SubOpt}^{\ast}_{\mathrm{CTL}}(\beta^{\ast}, k, \epsilon, \alpha, N) \geq \Omega\left(\alpha^{1-1/k} + \left(\frac{1}{\epsilon} \sqrt{\frac{\beta^{\ast}}{N}}\right)^{1-1/k}\right)$
>
>
>
>
> **Technical flaw in Tao et al. 2022.** Please see our general response above.

---

> > ### Comment · Reviewer_1jNj · 2024-08-12
> >
> > Thank you for your response, I will keep my score positive.

---

> > > ### Author Response · Authors · 2024-08-13
> > > **Thank you**
> > >
> > > Thanks again for your comments and positive evaluation of our paper!

---

### Author Rebuttal · Authors · 2024-08-05

# Technical flaw in Tao et al. 2022

We thank all the reviewers for their time and insightful comments. Since all reviewers would like to see more discussion of Tao et al. 2022's technical flaw from the perspective of upper bound proof, we will provide a general global response below.

The key flaw in their upper bound proof is **the incorrect bound on the number of pulls for each sub-optimal arm** (in the proof of their Lemma 13).

This step happens on Page 24 of their paper (see "Lastly, for any fixed sub-optimal arm a..."). It should be $\Delta_a \ge D_{\tau(a)}$, rather than $\Delta_a \ge D_{\tau(a)}^2$. After fixing this, one gets the correct bound on the number of pulls for each sub-optimal arm $a$ as (focus on key terms of $\epsilon$ and $\Delta_a$ only)
$${O}\left(\frac{1}{\epsilon^2 \Delta_a^{\frac{2(1+v)}{v}}}\right)$$, rather than the wrong one in their proof, which has $\Delta_a^{\frac{1+v}{v}}$ (i.e., missing the square).

In fact, translating this using our notation (i.e., $k = 1+v$), the correct bound above becomes $${O}\left(\frac{1}{\epsilon^2 \Delta_a^{\frac{2k}{k-1}}}\right)$$, which is exactly the first term in Eq. (9) of our paper.

Thus, following the same proof step in our paper (ignore all corruption terms), one can fix their proof and obtain the correct upper bound.

**Remark.** We remark that in Tao et al.  2022: (i) the authors did not provide detailed proof for the problem-independent (minimax) bound. They only provided the proof for the problem-dependent one (which is also ungrounded, as discussed above). Then, based on this, the authors claimed to obtain the problem-independent one (with standard techniques); (ii) Tao et al. 2022 follow an arm-elimination approach while our analysis is based on UCB. Nevertheless, it is well-known that in both approaches, the key step is to upper bound the number of pulls for each sub-optimal arm.

---

### Decision · Program_Chairs · 2024-09-25

**Decision:**

Accept (poster)

**Comment:**

The paper provides novel characterization for local differential private and robust multi-armed bandits against Huber corruption. It also fixes a technical flaw in previous works.  All reviewers agree that the paper is interesting and makes descent contribution to the field.